# Natural variation of BSK3 tunes brassinosteroid signaling to regulate root foraging under low nitrogen

Zhongtao Jia [1], Ricardo F.H. Giehl [1], Rhonda C. Meyer [2], Thomas Altmann [2] & Nicolaus von Wirén [1]

Developmental plasticity of root system architecture is crucial for plant performance in nutrient-poor soils. Roots of plants grown under mild nitrogen (N) deficiency show a foraging response characterized by increased root length but mechanisms underlying this developmental plasticity are still elusive. By employing natural variation in Arabidopsis accessions, we show that the brassinosteroid (BR) signaling kinase BSK3 modulates root elongation under mild N deficiency. In particular, a proline to leucine substitution in the predicted kinase domain of BSK3 enhances BR sensitivity and signaling to increase the extent of root elongation. We further show that low N specifically upregulates transcript levels of the BR co-receptor *BAK1* to activate BR signaling and stimulate root elongation. Altogether, our results uncover a role of BR signaling in root elongation under low N. The BSK3 alleles identified here provide targets for improving root growth of crops growing under limited N conditions.

[1] Molecular Plant Nutrition, Leibniz Institute of Plant Genetics and Crop Plant Research (IPK), Corrensstrasse 3, 06466 Gatersleben, Germany. [2] Heterosis, Leibniz Institute of Plant Genetics and Crop Plant Research (IPK), Corrensstrasse 3, 06466 Gatersleben, Germany. Correspondence and requests for materials should be addressed to N.Wirén. (email: vonwiren@ipk-gatersleben.de)

In quantitative terms, nitrogen (N) is the mineral element required in largest amounts by plants. However, soils in natural or agro-ecosystems often do not provide sufficient levels of N to sustain optimal plant growth and development. As soil N constantly undergoes transformation processes, pool sizes of different inorganic and organic N forms change frequently, so that plant-available N pools can vary largely over time and within very short horizontal and vertical distances[1]. Thus, to optimize access to N under limiting conditions, root systems must continuously sense and respond to local or temporal fluctuations in N availability[2–4].

Previous experiments with *Arabidopsis thaliana* have shown that specific root architectural modifications can be induced by nutrient-derived signals that act locally[5–8] or systemically[9,10]. In growth substrates with heterogeneous N availability, plant roots preferentially colonize N-enriched patches by targeted lateral root development. Whereas nitrate ($NO_3^-$) mainly stimulates lateral root elongation[5,6,11], ammonium ($NH_4^+$) induces lateral root branching[7], supporting the view that these two major inorganic N forms shape root system architecture in a complementary manner. When N is evenly distributed in the substrate, root system architecture responds to a limiting dose of N in a dual manner[9]. Being exposed to very low external N, plants adopt a "survival strategy," in which the elongation of both primary and lateral roots, as well as the emergence of new lateral roots is inhibited[9,12]. Besides the involvement of NRT1.1-dependent auxin removal from lateral root primordia[13], this root architectural modification also depends on a regulatory module consisting of CLE-type signaling peptides and their receptor protein CLV1[14,15]. In N-deficient roots, *CLE1*, *CLE3*, *CLE4*, and *CLE7* are upregulated and their corresponding peptides are suggested to move from root pericycle cells to phloem companion cells, where they interact with CLV1 to inhibit the outgrowth and emergence of lateral roots[14].

In contrast to severe N limitation, external N levels that induce only mild deficiency stimulate the emergence of lateral roots[16] and especially the elongation of primary and lateral roots[9,12]. Although this stimulatory response is of particular interest as it reflects a "systemic foraging strategy" that increases the soil volume explored by the root system, it is the least understood N-dependent architectural adjustment. Also here auxin appears to play an active role, as the auxin biosynthesis gene *TAR2* is upregulated by low N and the *tar2* mutant displays inhibited lateral root emergence under mild N deficiency[16]. However, as the length of primary and lateral roots in *tar2* mutants remained unaffected, TAR2-dependent auxin biosynthesis alone cannot explain how mild N deficiency stimulates root elongation.

Here we assess the natural variation in root growth under mild N deficiency in 200 accessions of *A. thaliana*. By performing a genome-wide association study (GWAS), we identify BSK3 (brassinosteroid signaling kinase 3), which mediates brassinosteroid (BR) signaling downstream of the BR receptors, as a major determinant for primary root length adjustment to low N. Furthermore, we demonstrate that low N activates BR signaling by upregulating the transcription of *BAK1*, and that allelic variation of BSK3 coordinates the signaling amplitude that tunes the root growth to external N availability.

## Results

### GWAS maps primary root length variation at low N to BSK3.
We set out to identify genetic components that modulate root growth under mild N deficiency by genome-wide association (GWA) mapping. For this purpose, we used a diverse panel of 200 accessions of *A. thaliana* reflecting a wide geographic distribution (Supplementary Fig. 1). After 1 week of pre-culture with sufficient

N, plants were transferred to either 11.4 mM N (high N, HN) or 0.55 mM N (low N, LN), a concentration that induces a strong systemic root foraging response in the accession Col-0[9]. After 9 days on treatments, we measured the primary root length of all accessions and observed a high degree of natural variation with primary root lengths ranging from 3 to 10.8 cm at HN and from 3.5 to 12.5 cm at LN (Fig. 1a, Supplementary Data 1). On average, primary roots of all examined accessions were 16% longer at LN than at HN ($P < 2.2e − 16$; Fig. 1a). The broad-sense heritability ($h^2$) for primary root length was estimated to be 88.8% and 85.7% for HN and LN, respectively.

To uncover genetic loci associated with the variation of primary root length, we performed GWA analysis using a mixed model that corrects for population structure[17]. At LN, we mapped two major loci above a threshold of 10% false discovery rate (FDR) on chromosomes 3 and 4 (Fig. 1b, Supplementary Data 2). The associated single-nucleotide polymorphism (SNP) on chromosome 3, which accounted for 6.3% of the natural variation for primary root length at LN, was located in *PHOSPHATE 1* (*PHO1*), a gene modulating root allometry in response to auxin, ABA, and nitrate[18]. Compared with wild type, the *pho1-2* mutant failed to stimulate primary root elongation under LN (Supplementary Fig. 2), supporting the hypothesis that *PHO1* was the underlying gene for this locus. The locus on chromosome 4 contained 13 SNPs (FDR < 0.1) and the most significantly associated SNP, which explained 11.7% of the observed phenotypic variation, was located at position 386,519 (Fig.1b, Supplementary Data 2). To further resolve the multiple SNPs associated with this locus, we employed multi-locus mixed model (MLMM)[19], which uses a stepwise model selection. The optimal model selected by this method identified the only SNP found in our initial GWAS on chromosome 4 (Supplementary Fig. 3). Estimates of the linkage disequilibrium (LD) between the top GWA SNP with surrounding markers revealed the presence of a long-range LD pattern, which may account for the multiple significantly associated SNPs identified in this region. We assumed a confidence interval by computing the LD with surrounding markers ($r^2 > 0.7$) starting from 292,979 to 398,078, a region that includes 31 genes (Supplementary Data 3). For 16 of these genes, we could obtain T-DNA insertion lines that we phenotyped to identify the causal gene underlying this locus. The top SNP was located in the gene *ER-TYPE $Ca^{2+}$-ATPASE 2* (*ECA2*; AT4G00900). However, primary root length at both N conditions was similar to wild type in two independent *eca2* T-DNA insertion lines (Fig. 1c). An insertion mutant in the GSK3/Shaggy-like kinase gene *ASKTHETA* (*ASKΘ*; AT4G00720) had shorter primary roots than its respective wild type, irrespective of the N treatment (Fig. 1d). Interestingly, an insertion mutant of *BSK3* (AT4G00710) showed a conditional phenotype, with a significantly shorter primary root than the wild type at LN, but not at HN (Fig. 1e). The primary root lengths of the remaining insertion lines were indistinguishable from wild type irrespective of the N condition (Supplementary Fig. 4).

To further ascertain the causal gene for the associated locus, we assessed the expression of *ASKΘ* and *BSK3* in nine accessions that showed contrasting primary root lengths (Supplementary Fig. 5). Transcript levels of *ASKΘ* and *BSK3* did not correlate significantly with primary root length of the tested accessions in either N environment (Fig. 1f, g), suggesting that either these genes are not causal for the identified marker-trait association or that polymorphism in the coding region contributes to the observed phenotypic variation. Therefore, we analyzed the coding sequences (CDS) of 139 re-sequenced accessions and searched for SNPs that could lead to changes in protein sequence. Whereas only one synonymous substitution (C774G) was detected in the CDS of *ASKΘ*, two synonymous (G1353A and A1413G) and one

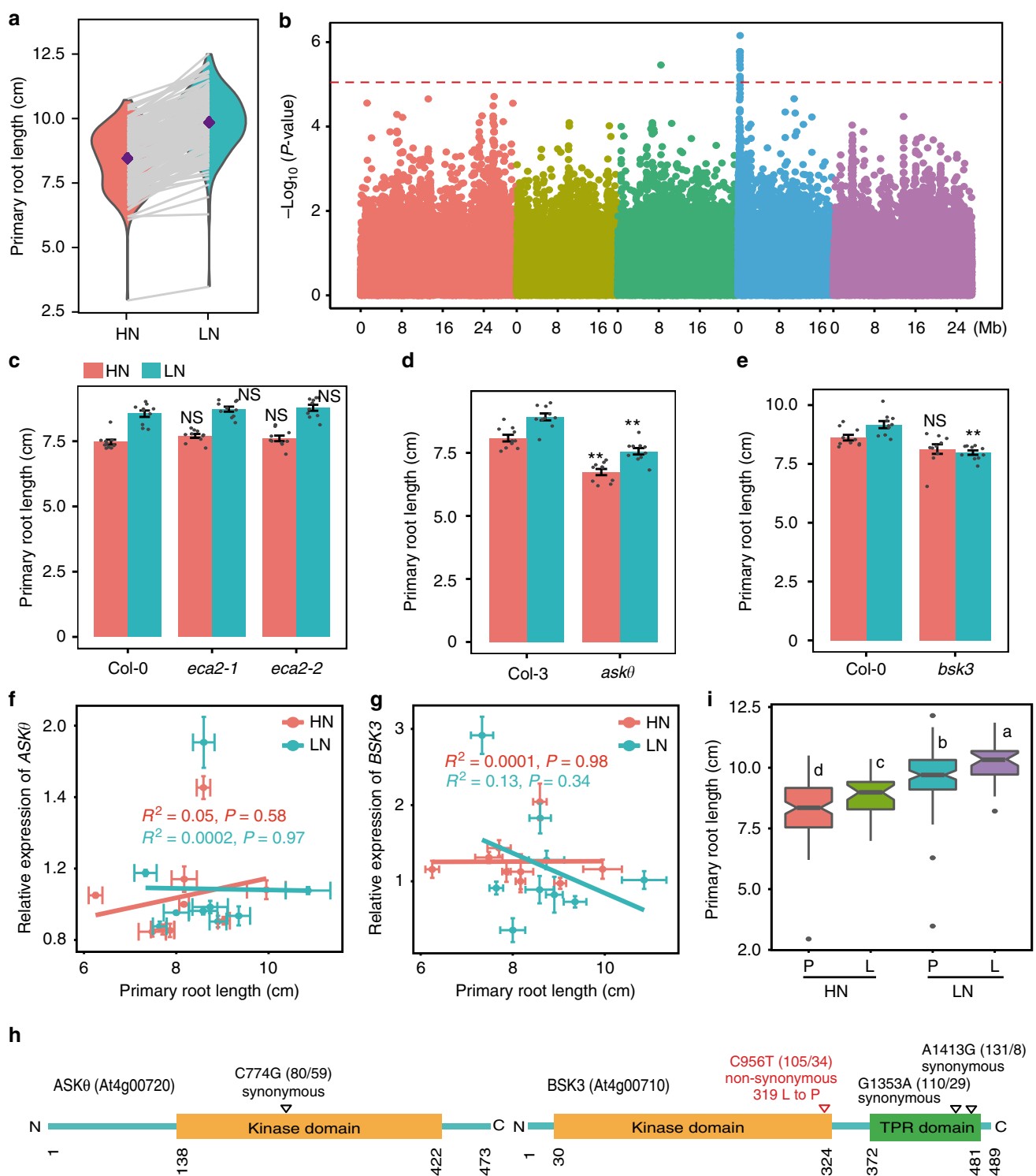

non-synonymous (C956T) substitutions were found in the CDS of *BSK3* (Fig. 1h, Supplementary Table 1). This non-synonymous mutation results in a leucine (L) to proline (P) substitution in the predicted N-terminal kinase domain of the protein. A protein haplotype analysis of BSK3 showed that accessions with the BSK3-L allele had longer primary roots than those accessions carrying the BSK3-P allele (Fig. 1i), suggesting that *BSK3* rather than *ASKΘ* is the causal gene for this locus. In support to this assumption, we also found that the likely functional SNP (i.e., C956T) was in strong LD with the GWA SNP ($r^2 = 0.73$), and that G1353A showed a moderate association ($-\log_{10} P$-value =

4.1) with primary root length under LN. Although we cannot fully rule out the contribution of other genes at this locus, these independent approaches strongly indicated that *BSK3* is the causal gene for the observed marker-trait association found on chromosome 4 (Fig.1b).

**BSK3 and its homologs modulate root growth at low N**. Analysis of the root system architecture demonstrated that the shorter primary root length of *bsk3* plants was associated to their inability to stimulate root elongation in response to mild N deficiency

**Fig. 1** Genetic variation and GWAS of primary root length grown under low N. **a** Reaction norms and phenotypic variation of primary root length of 200 natural accessions of *A. thaliana* grown under HN vs. LN for 9 days. Purple diamonds represent means of primary root length for 200 accessions under each N treatment. **b** Manhattan plot for the SNP associations to primary root length under LN. Negative log10-transformed *P*-values from a genome-wide scan were plotted against positions on each of the five chromosomes of *A. thaliana*. Chromosomes are depicted in different colors (I–V, from left to right). The red dashed line corresponds to the Benjamini and Hochberg false discovery rate level of *q* < 0.1. **c–e** Primary root length of T-DNA knockout lines for *ECA2* (**c**), *ASKθ* (**d**), and *BSK3* (**e**) grown under two N conditions for 9 days. Bars represent means ± SEM (*n* = 10 independent biological replicates). Asterisks indicate statistically significant differences to wild type according to Welch's *t*-test (**P* < 0.01; NS, not significant). **f, g** Correlation of *ASKθ* (**f**) or *BSK3* (**g**) transcript levels in roots with primary root length at either HN or LN. **h** Schematic representation of ASKθ and BSK3 protein sequences highlighting relevant protein domains. TPR, tetratricopeptide. Location, nucleotide polymorphism, and effect at the amino acid level for each identified SNP are shown. Numbers in brackets denote the number of lines carrying the corresponding allele. (**i**) Primary root length of natural accessions representing two BSK3 protein haplotypes (*n* = 105 and 34 accessions for P and L haplotypes, respectively). Horizontal lines show medians; box limits indicate the 25th and 75th percentiles; whiskers extend to 5th and 95th percentiles. Different letters indicate significant differences at *P* < 0.05 according to one-way ANOVA and post-hoc Tukey's test

(Supplementary Fig. 6a, b). Furthermore, compared with wild-type plants, *bsk3* plants had 13% shorter lateral roots but no differences in lateral root density at LN (Supplementary Fig. 6c, e). In consequence of failed stimulation of primary root elongation and partial loss of lateral root length under mild N deficiency, total root length of *bsk3* plants was only 78% of that of wild-type plants (Supplementary Fig. 6d). Compared with *bsk3*, the elongation of both primary and lateral roots in response to LN could be significantly increased by introducing a genomic fragment containing the promoter and coding regions of *BSK3* into the *bsk3* mutant (Supplementary Fig. 6).

Previously, it has been reported that BSK3 functions in the BR signaling cascade[20–22]. We found that both primary and lateral roots of N-deficient plants were hypersensitive to exogenous supply of the bioactive BR 24-epibrassinolide (BL) (Supplementary Fig. 7a–c). Furthermore, supply of the BR biosynthesis inhibitor brassinazole largely prevented the low N-induced elongation of primary and lateral roots (Supplementary Fig. 7d–f), suggesting that BRs play a role in root system architectural changes under LN. It has been shown that BSK3 acts redundantly with other BSK family members in transducing the BR signal from the plasma membrane to the cytosol[21]. To further assess the role of BSK-dependent BR signaling in root architectural changes under mild N deficiency, we analyzed double, triple, and quadruple mutants for *BSK3*, *BSK4*, *BSK7*, and *BSK8*. In line with the partially redundant function of these most closely related BSKs[21], the *bsk3,4,7,8* quadruple mutant exhibited the strongest decrease in root length as compared with the *bsk3* single mutant at both N conditions (Fig. 2a–c). Relative to *bsk3*, only minor effects were observed for *bsk3,4* double and *bsk3,4,7* triple mutants at HN condition, whereas no effect was observed for *bsk3,4* double and *bsk3,4,7* or *bsk3,4,8* triple mutants at LN. Notably, the primary root length in response to LN was similarly attenuated in all tested mutants, except for *bsk3,4,7* (Fig. 2a, b). The additive effect of all four BSKs was particularly relevant for average lateral root length, as *bsk3,4,7,8* plants failed to significantly elongate lateral roots at LN (Fig. 2c). Consequently, these plants showed no significant increase in total root length under LN supply (Supplementary Fig. 8a). Relative to Col-0, none of the tested mutants showed significant changes in lateral root density (Supplementary Fig. 8b).

Microscopic analyses revealed that the primary root response to LN of Col-0 plants was associated with enhanced mature cell length rather than altered meristem size (Fig. 2d–f). Accordingly, the attenuated primary root response of *bsk3* and *bsk3,4,7,8* mutants to LN was associated with a reduced ability of these plants to stimulate mature cell elongation (Fig. 2d–f). Altogether, these results indicate that BSK3-mediated BR signaling plays a critical role in modulating cell elongation and, consequently, root elongation in response to LN. Our data also demonstrate that this

function is, at least in part, functionally compensated by BSK4, BSK7, and BSK8.

**BSK3 variants modulate root elongation and BR sensitivity.** Although our initial data indicated that an L to P substitution in the position 319 of the predicted kinase domain of BSK3 contributes to the natural variation of primary root length under LN (Fig. 1h, i), this evidence remained mainly based on correlations. Therefore, we used an allelic swapping approach and transformed the *bsk3,4,7,8* mutant with sequences coding for the P or L variant. Both BSK3 alleles recovered primary root growth when expressed under the control of the *BSK3*Col-0 promoter (Fig. 3a, b). However, transgenic lines carrying the BSK3-L allele exhibited significantly longer primary roots than those complemented with the BSK3-P allele (Fig. 3a, b). To further test whether the L to P substitution affects BR signaling, we assessed plant responses to BL. As expected[20–22], the exogenous supply of BL inhibited primary root length by 65% and stimulated hypocotyl elongation by 160% in wild-type plants (Fig. 4a–c). These responses were strongly attenuated in the *bsk3,4,7,8* quadruple mutant. Although both BSK3 alleles could partially rescue the BR responsiveness of *bsk3,4,7,8* plants, expression of BSK3-L restored BR sensitivity to a larger extent than BSK3-P (Fig. 4b, c). To control for variation at the protein level, we produced translational fusions between the two BSK3 protein variants and green fluorescent protein (GFP), and introduced them into *bsk3,4,7,8* plants. Both variants recovered BR sensitivity, but complementation was more efficient with BSK3-L-GFP irrespective of variations in GFP fluorescence intensity (Supplementary Fig. 9). Further analysis of BR sensitivity in 56 natural accessions revealed that accessions carrying the BSK3-L variant were more sensitive to exogenous supply of BR than those with BSK3-P (Fig. 4d, e, Supplementary Data 4). Taken together, these results showed that BSK3-L is more efficient than BSK3-P in mediating BR signaling.

Previously, it has been shown that root elongation can be impaired by genetically enhancing BR signaling[23]. Thus, we overexpressed the two BSK3 variants in the *bsk3,4,7,8* background using the CaMV *35S* promoter. Although both BSK3 variants partially recovered root growth of *bsk3,4,7,8* (Fig. 4f, g), transgenic lines overexpressing BSK3-L showed significantly shorter and slower growing primary roots than those expressing BSK3-P (Fig. 4g, h). In addition, transgenic lines overexpressing BSK3-L exhibited stronger BR sensitivity than those overexpressing BSK3-P (Supplementary Fig. 10). These results further demonstrated that the BSK3-L allele relays a stronger BR signaling output than the BSK3-P allele.

Our initial expression analysis indicated that polymorphisms affecting the transcript levels of *BSK3* are unlikely causal for the observed natural variation of root growth (Fig. 1g). To verify this

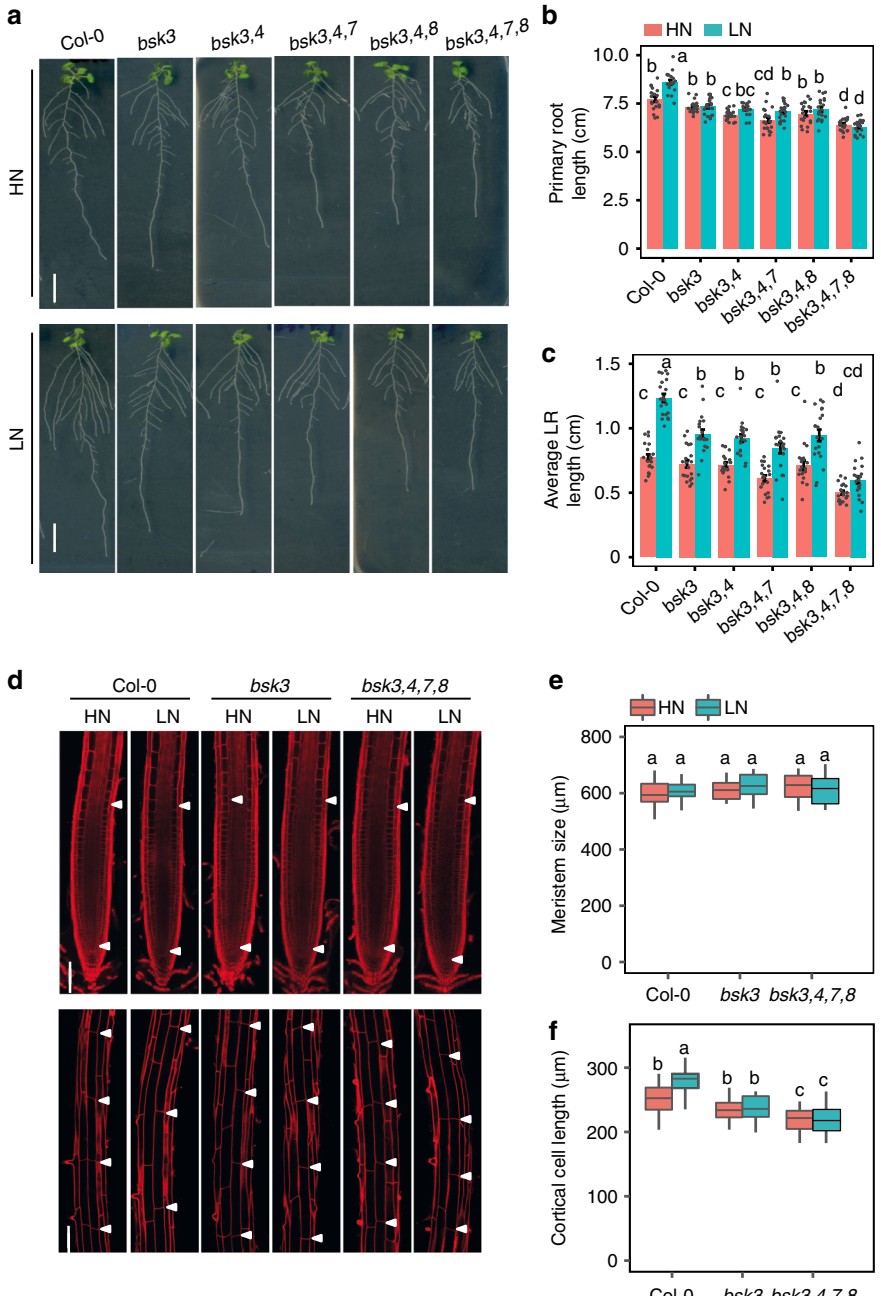

**Fig. 2** Root architecture of *bsk3* and multiple *bsk* knockouts under low N. **a–c** Appearance of plants (**a**), primary root length (**b**), and average lateral root length (**c**) of wild-type (Col-0) and single or multiple *bsk* mutant plants grown at HN or LN. Bars represent means ± SEM ($n = 20$ independent biological replicates). Scale bars, 1 cm. **d** Representative confocal images of root meristem (upper panel) and length of cortical cells (bottom panel) of wild-type (Col-0), *bsk3*, and *bsk3,4,7,8* grown under HN or LN. In the upper panel, white arrowheads indicate the position of quiescent center (QC) and the boundary between the meristematic zone and elongation zone, whereas in the bottom panel they indicate the boundaries of two consecutive cortical cells. Scale bars, 100 μm. **e**, **f** Length of meristem (**e**) and mature cortical cells (**f**) of wild-type (Col-0), *bsk3*, and *bsk3,4,7,8* grown under HN or LN ($n = 15$ independent biological replicates). Root system architecture and cellular traits were assessed after 9 days. Horizontal lines show medians; box limits indicate the 25th and 75th percentiles; whiskers extend to 5th and 95th percentiles. Different letters indicate significant differences at $P < 0.05$ according to one-way ANOVA and post-hoc Tukey's test

assumption, we expressed in *bsk3,4,7,8* the CDS of L-type *BSK3* under the control of the *BSK3* promoter from either Cvi-0 or Col-0, two accessions that differ in primary root length under LN (Supplementary Fig. 11a). If putative differences in *BSK3* expression determine the phenotypic variation between Cvi-0 and Col-0, we expected to detect different complementation efficiencies. However, primary root growth (Supplementary Fig. 11b, c) and in particular BR sensitivity (Supplementary

Fig. 12) were recovered to a similar extent in transgenic lines expressing either construct. We therefore concluded that variation in the promoter sequence of *BSK3* is not causally associated with the natural variation of root growth and BR sensitivity.

**BSK3-dependent BR signaling tunes root responses to low N.** As natural allelic variations in BSK3 modulated BR sensitivity and signaling, we tested whether these variations also determine the

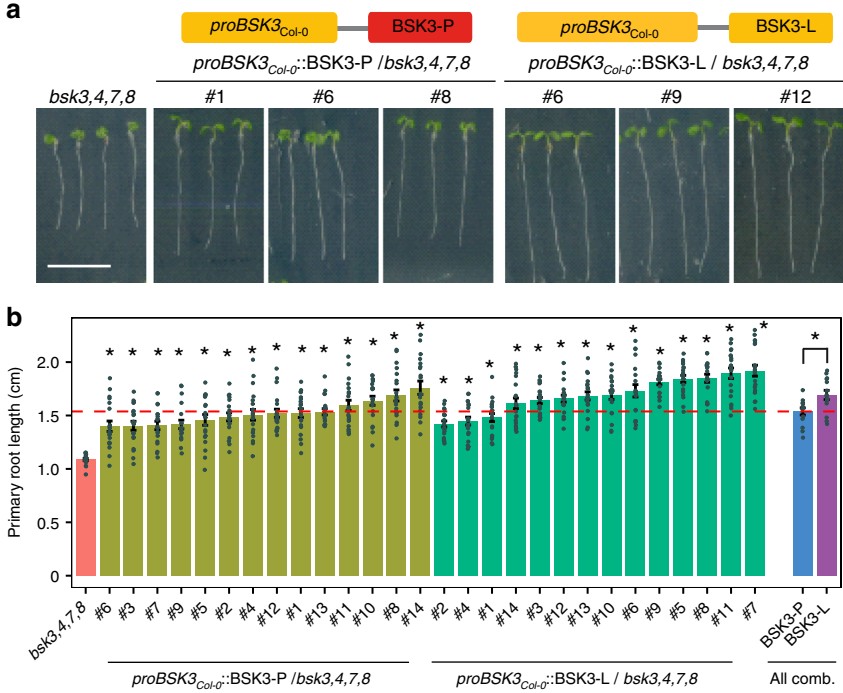

**Fig. 3** Allelic variation of BSK3 determines root length. **a** Schematic of transgenic constructs used for complementation of the *bsk3,4,7,8* quadruple mutant and representative photographs of 6-day-old plants. Three representative lines for each construct are shown. Scale bar, 1 cm. **b** Primary root length of *bsk3,4,7,8* and 14 independent T2 lines for each construct. Bars represent means ± SEM ($n = 20$ independent biological replicates). Average values of 14 independent lines for each construct are also shown (all comb.). Asterisks indicate statistically significant differences between complementation lines and *bsk3,4,7,8* mutant or between two BSK3 protein variants according to Welch's *t*-test (*$P < 0.01$)

root responsiveness to mild N deficiency. As *bsk3,4,7,8* plants exhibit short primary and lateral roots already when grown on HN (Fig. 2a–c), relative root lengths (i.e., their ratio under LN to HN) were calculated to allow for direct comparison. In line with our initial experiments (Fig. 2), the primary root of the *bsk3,4,7,8* quadruple mutant was largely insensitive to LN due to attenuated cell elongation (Fig. 5a, e–g). Although complementation with BSK3-P only partially recovered the responsiveness of *bsk3,4,7,8* to LN, BSK3-L was able to fully rescue this response by significantly increasing cell elongation without altering meristem length (Fig. 5a, e–g). These results suggest that the BSK3-L variant increases the responsiveness of the primary root to LN. BSK3-L also restored more efficiently average lateral root length and total root length, whereas none of the BSK3 alleles could recover the suppressed lateral root density of *bsk3,4,7,8* (Fig. 5b–d).

We then tested whether the superior root length brought about by the L-allele also led to elevated N uptake. Although under LN supply *bsk3,4,7,8* accumulated significantly less N in the shoot, complementation with either BSK3 variant did not restore shoot N accumulation (Supplementary Fig. 13). This failure may be due to additional, non-redundant functions of BSK proteins and of BRs in the transcriptional regulation of N transporters[24,25]. However, significant correlations were found between BR sensitivity, as determined by root length inhibition or hypocotyl elongation, and the responsiveness of primary root or average lateral root length to mild N deficiency (Fig. 5h, i, Supplementary Fig. 14), indicating a genetically determined coupling of BR response and root response to LN. Together, these data pointed to a specific role of BSK3 in root elongation and demonstrated that the BR sensitivity gained by L319P substitution in BSK3 is critical for the extent of root elongation under LN. Thus, we concluded that BSK3-dependent modulation of BR signaling

contributes significantly to the natural variation in root length responses to N availability.

**Localization and N-dependent regulation of *BSK3* expression.** As our experiments with *BSK3* insertion and complementation lines revealed N-dependent root growth phenotypes, we checked *BSK3* expression in whole roots of Col-0 plants grown for 9 days on HN or LN and found no significant alteration (Fig. 6a). To investigate whether putative expression changes are confined to particular root zones, we generated transgenic lines expressing GUS or a BSK3-GFP translational fusion under control of the *BSK3* promoter from Col-0. Nine days after transferring plants to HN or LN, GUS activity was detected in all root tissues of both primary root and lateral roots (Fig. 6b, Supplementary Figs. 15 and 16). Confocal microscopy of plants expressing *proBSK3::BSK3-GFP* further revealed that the protein was present from the meristematic zone all the way through the elongation zone of epidermal, cortical, and endodermal cells (Fig. 6c). Confined localization at the cell border was in agreement with the recently confirmed binding of the myristoylated protein to the plasma membrane[22]. External N supply had no effect on the pattern or intensity of *BSK3* expression. Moreover, a time-course histological staining and expression analysis in roots did not reveal any significant differences in *BSK3* expression under prolonged N deficiency (Supplementary Fig. 15). We further monitored *BSK3* promoter activity and protein abundance at different stages of lateral root development[26]. Under both N conditions, *BSK3* promoter activity and BSK3-GFP fluorescence were only detected in lateral roots that had emerged from parental roots (Fig. 6d, Supplementary Fig. 16). Overall, these data indicate that although *BSK3* expression is not regulated by N, it coincides spatially with developmental stages associated with cell elongation in roots.

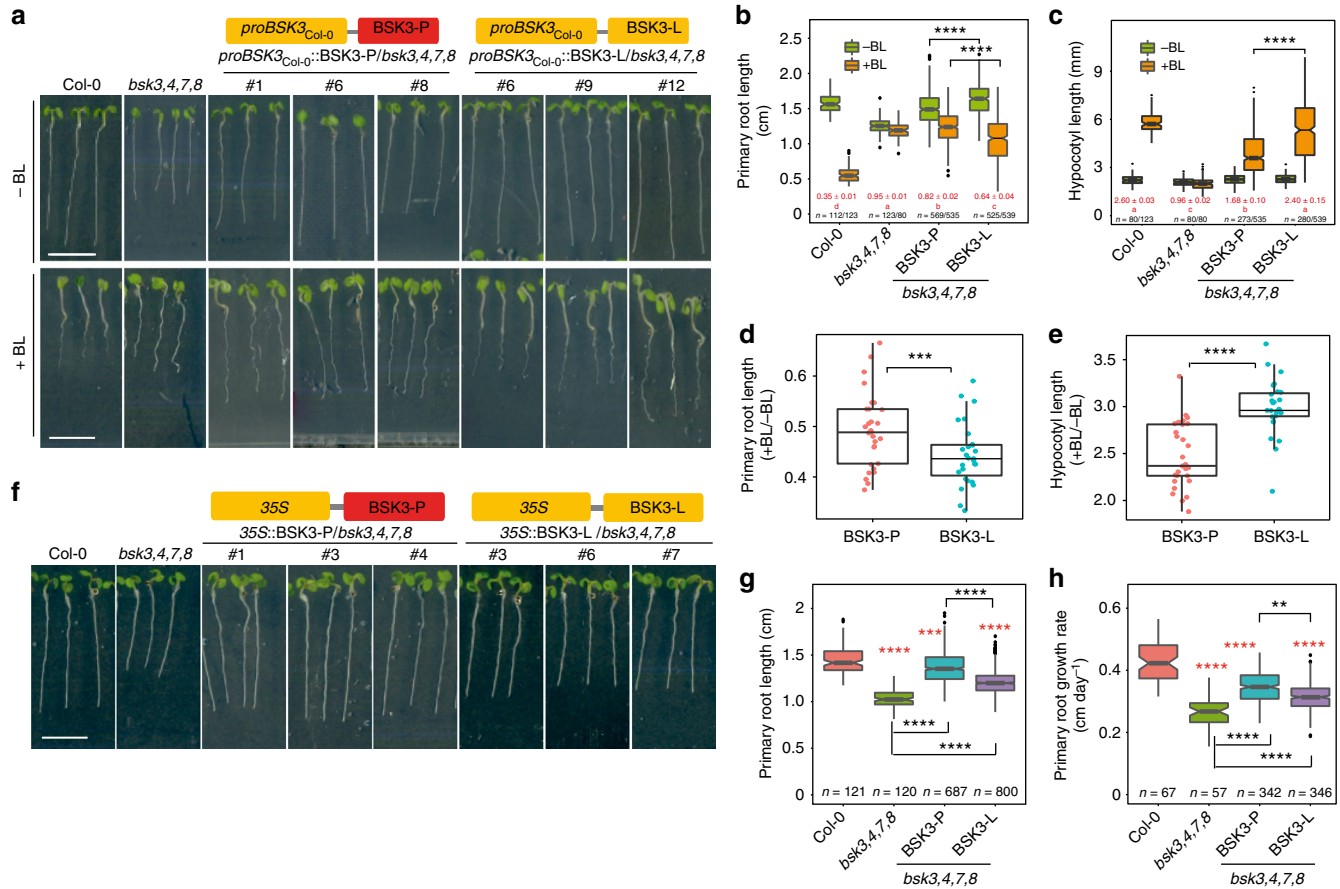

**Fig. 4** Allelic variation of BSK3 determines plant sensitivity to BR. **a–c** Brassinosteroid sensitivity of wild-type (Col-0), *bsk3,4,7,8*, and independent transgenic lines expressing sequences coding for either the BSK3-P or BSK3-L variant under the control of the *BSK3*<sub>Col-0</sub> promoter. Appearance (**a**), primary root length (**b**), and hypocotyl length (**c**) of 6-day-old plants. Numbers in red indicate relative root or hypocotyl length (+BL/−BL). Different letters indicate significant differences for relative root or hypocotyl length at $P < 0.05$ according to one-way ANOVA and post-hoc Tukey's test. **d, e** Primary root (**d**) and hypocotyl (**e**) growth responses of natural accessions carrying BSK3-P or BSK3-L to the exogenous supply of 1 μM 24-epibrassinolide ($n = 29$ and 27 accessions for P and L haplotypes, respectively). **f–h** Root growth of wild-type (Col-0), *bsk3,4,7,8*, and independent transgenic lines expressing sequences coding for either the BSK3-P or BSK3-L variant under the control of the CaMV *35 S* promoter. Appearance (**f**), primary root length (**g**), and root growth rate (**h**) of 6-day-old plants. Horizontal lines show medians; box limits indicate the 25th and 75th percentiles; whiskers extend to 5th and 95th percentiles. Asterisks indicate statistically significant differences to wild-type (in red) or to *bsk3,4,7,8* (in black), or between two BSK3 protein haplotypes (in black) according to Welch's *t*-test (**$P < 0.01$; ***$P < 0.001$; ****$P < 0.0001$). Scale bars, 1 cm

**N deficiency promotes BR signaling by upregulating *BAK1*.** In Arabidopsis, BSK3 mediates BR signaling downstream of the BR receptors[20,27]. As *BSK3* expression is not regulated by N (Fig. 6), we hypothesized that an N-dependent signal might enter BR signaling upstream of BSK3. Therefore, we conducted a time-course expression analysis to assess whether root expression of BR receptors responds to external N. After 6 days of growth on conditions that evoke mild N deficiency, *BAK1* transcript levels were significantly upregulated (Fig. 7a). Notably, the response of primary and lateral roots to mild N deficiency was significantly attenuated in *bak1-1* mutant plants (Fig. 7b–d). Furthermore, similar to *bsk3*, *bak1-1* plants also failed to induce cell elongation in response to LN (Supplementary Fig. 17). As BAK1 is involved in multiple signaling pathways[28], we tested whether impaired root growth of *bak1-1* plants was indeed due to perturbed BR signaling. Ectopic expression of *BSK3* was able to largely restore the sensitivity of *bak1-1* plants to exogenous BR (Supplementary Fig. 18), as well as their primary and lateral root growth under LN (Fig. 7e–g), corroborating that the reduced root response of *bak1-1* plants to LN is due to impaired BR signaling. Unlike *BAK1*, expression of *BRI1* was not regulated by LN (Supplementary Fig. 19a). Although exhibiting shorter primary and lateral roots compared with wild-type plants under both N conditions, two independent *bri1* loss-of-function mutants were still able to stimulate primary root and lateral root growth under LN to a similar extent as the wild type (Supplementary Fig. 19b–f). These results indicate that mild N deficiency activates BR signaling upstream of BSK3 by upregulating *BAK1* expression.

## Discussion

Root developmental plasticity is crucial for optimizing nutrient capture in continuously changing or nutrient-depleted soil environments. Several root architectural responses have been related to changes in phytohormone balance or fluxes. In particular, altered auxin transport or signaling has been shown to mediate adaptive responses of roots to variable nutrient supplies[8,13,29–32]. To uncover genetic components that modulate root architectural plasticity to N availability, we phenotyped the root response of *A. thaliana* accessions to low N and identified the BR signaling gene *BSK3* being causal for the natural variation of primary root length under altered N availability. These results reveal a previously unknown role of BRs in shaping root system architecture in response to mild N deficiency.

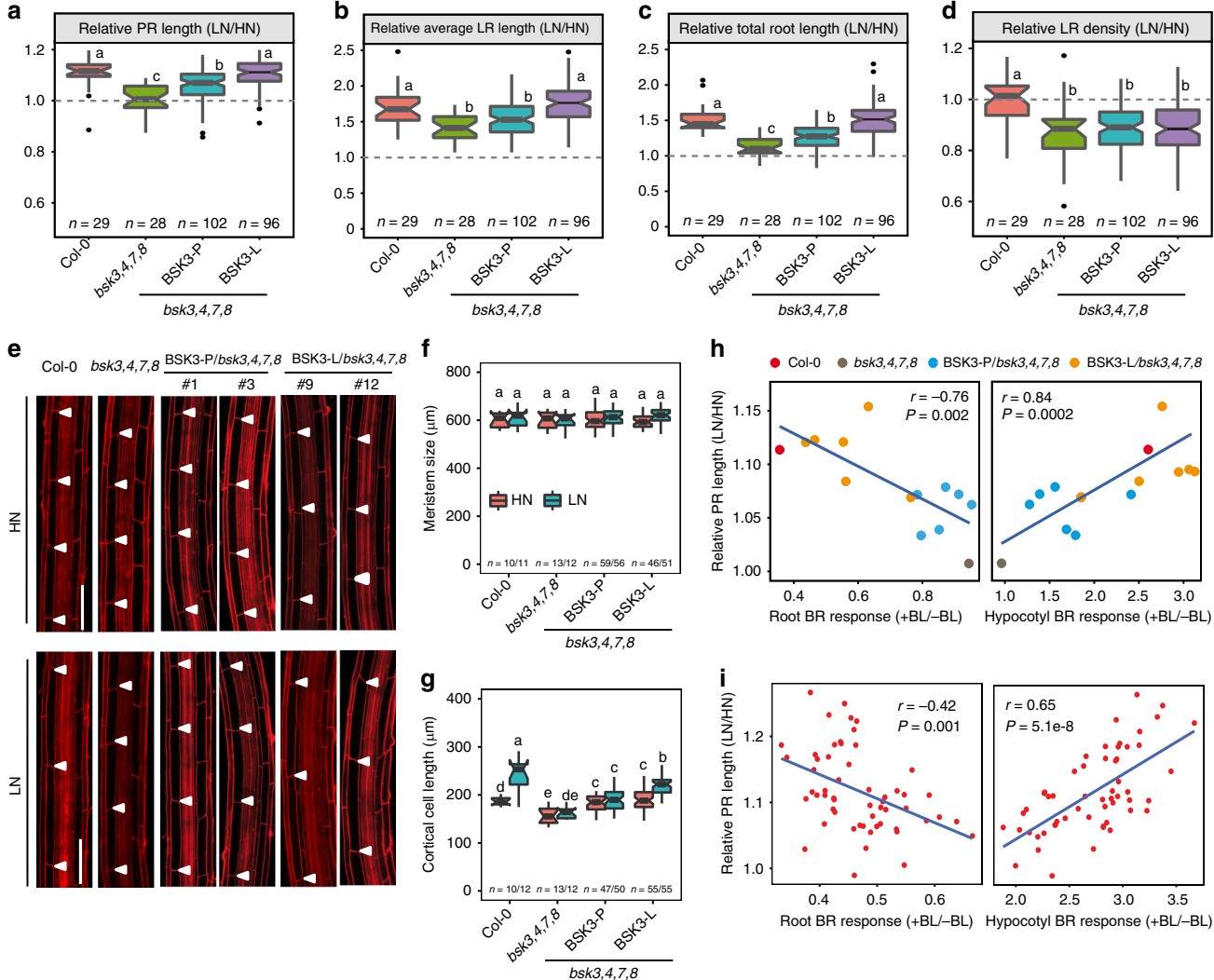

**Fig. 5** BSK3-dependent BR sensitivity tunes root responses to low N. **a–d** Response to low N of Col-0, *bsk3,4,7,8*, and independent transgenic plants expressing the sequences coding for either the BSK3-P or BSK3-L variant under control of the *BSK3*<sub>Col-O</sub> promoter. Relative change of primary root length (**a**), average lateral root length (**b**), total root length (**c**), and lateral root density (**d**) in response to low N. **e** Representative confocal images of mature cortical cells of wild-type (Col-0), *bsk3,4,7,8* transgenic lines complemented with either BSK3 variants under *BSK3*<sub>Col-O</sub> promoter (two representative lines for each construct are shown). White arrowheads indicate two consecutive cortical cells. Scale bars, 100 μm. **f, g** Length of meristems (**f**) and mature cortical cells (**g**) of wild-type (Col-0), *bsk3*, and allelic complementation lines in *bsk3,4,7,8* mutant grown under HN or LN. Root system architecture and cellular traits were assessed after 9 days. Horizontal lines show medians; box limits indicate the 25th and 75th percentiles; whiskers extend to 5th and 95th percentiles. Different letters indicate significant differences at *P* < 0.05 according to one-way ANOVA and post-hoc Tukey's test. **h, i** Correlation between BR-dependent root or hypocotyl elongation and the responsiveness to LN of primary root in BSK3 allelic variant complemented lines (**h**) and natural accessions (**i**). The data used for the correlation analyses derived from experiments shown in Fig. 1a, Fig. 4b–e and Fig. 5a

When exploring root architectural plasticity under deficiency of different nutrients, we observed that the Col-0 accession increases the length of primary and lateral roots under external concentrations that induce mild N deficiency[9]. Here we show that this foraging response is largely conserved in *A. thaliana*, as primary roots were significantly longer under mild N deficiency than at sufficient N in >95% of the 200 accessions tested (Fig. 1a, Supplementary Data 1). Based on considerable natural variation in primary root length at LN, we mapped by GWAS two genes, *PHO1* and *BSK3*, which explained 6.3% and 11.7% of the observed variation, respectively (Fig. 1 and Supplementary Figs. 2 and 6). Natural phenotypic variation can arise from polymorphisms in the *cis*-regulatory region, leading to altered transcript levels, or in the CDS, which can modify protein function[33–36]. In the case of *PHO1*, although located within an exon, the SNP did not result in amino acid exchange, suggesting

that *cis*-regulatory variation(s) at *PHO1* might drive natural variation of primary root length at LN. By contrast, in the case of BSK3, our results provide strong evidence that non-CDS variation at the *BSK3* locus is not responsible for the variation in primary root length under mild N deficiency, as (i) we found no correlation between *BSK3* transcript levels and primary root length (Fig. 1g), and (ii) promoters of accessions exhibiting differential response to LN complemented primary root length and BR sensitivity of *bsk3,4,7,8* at similar efficiency (Supplementary Figs. 11 and 12). This is in contrast with growth responses to severe N deficiency, in which root elongation correlated significantly with the expression levels of *JR1* and *PhzC* or of the signaling peptide *CLE3*[14,37]. Instead, we found one non-synonymous mutation (C956T) in the coding region of *BSK3* that causes a leucine to proline substitution at position 319 (Fig. 1h). As accessions with the L-allele had longer primary roots than

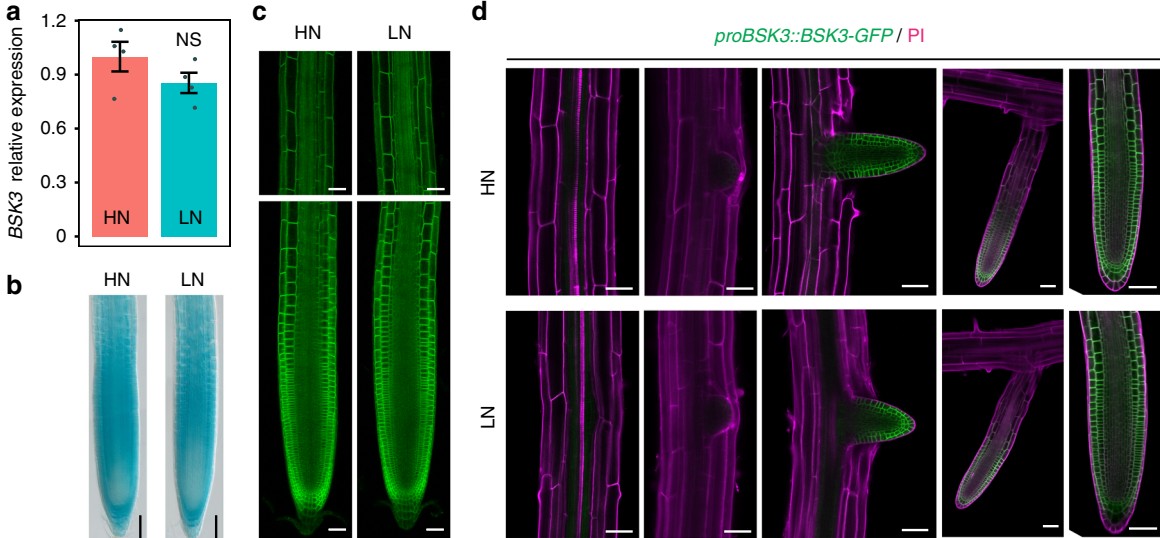

**Fig. 6** *BSK3* expression pattern and protein localization in response to N availability. **a** *BSK3* transcript levels in response to HN or LN. *BSK3* expression was assessed in whole roots by qPCR analysis and normalized to *ACT2* and *UBQ10*. Bars represent means ± SEM (*n* = 4 independent biological replicates). NS denotes no significant difference to HN according to Welch's *t*-test (*P* > 0.05). **b–d** *proBSK3*-dependent GUS activity and BSK3-GFP localization in the primary root (**b**, **c**) and lateral roots at several developmental stages (**d**). Samples were taken 9 days after transfer. Representative images (*n* = 15 biological replicates) are shown. PI, propidium iodide. Scale bars, 100 μm

those carrying the P-allele (Fig. 1i), we hypothesized that the C956T polymorphism in the CDS is the determining quantitative trait nucleotide underlying this phenotypic variation. Supporting this notion, we demonstrated that natural accessions carrying the BSK3-L variant or transgenic lines expressing BSK3-L in the *bsk3,4,7,8* quadruple mutant had stronger root growth and BR sensitivity than those expressing the BSK3-P variant (Figs. 3 and 4, Supplementary Figs. 9 and 10).

BSK3 is a plasma membrane-anchored protein belonging to a subfamily of 12 receptor-like cytoplasmic kinases[20,21]. Several BSKs, including BSK3, can interact with BR receptors upon BR binding[20–22,27]. For instance, upon BRI1-dependent phosphorylation BSKs become activated, are released from the BRI1/BAK1 receptor complex, and can interact with the Kelch-repeats-containing phosphatase BSU1[38,39], which in turn dephosphorylates and inactivates the GSK3/Shaggy-like kinases BIN2 and BIL2[40]. How BSKs activate BSU1 still remains unclear. All BSKs contain a putative N-terminal kinase domain and a C-terminal tetratricopeptide repeat motif[20]. However, consistent kinase activity has not been detected for any BSK protein[22,39,41,42], even though weak $Mn^{2+}$-dependent kinase activity was recorded in some instances[39,42]. It has therefore been proposed that the kinase activity of BSKs may rely on activation by another yet unknown binding partner. Alternatively, rather than being bona fide kinases, BSKs may function as scaffolds that facilitate the interaction between BSU1 and BIN2/BIL2[21,22,41]. Notably, mutating two invariable amino acids considered critical for the putative kinase activity of OsBSK3 made the protein non-functional[39]. Furthermore, it has been shown that missense mutations in the kinase domain of BSK3 result in loss-of-function or reduced BR responses[22]. However, the amino acid substitution found in our study does not relate to the phosphorylation site targeted by BRI1 or the residues assessed previously[22,39]. This is conceivable as alleles with deleterious consequences would likely not be present at a detectable frequency in natural accessions due to negative selection.

BRs regulate several physiological and developmental processes in plants. These plant steroid hormones have also been implicated in the transcriptional regulation of N uptake[24,25]. Regarding root growth, BRs modulate the elongation of

differentiated cells[43] and the size of meristems[23,44,45]. In particular, cell elongation is of relevance here, because primary root elongation under N deficiency was primarily associated with increased cortical cell size (Figs. 2d–f and 5e–g, Supplementary Fig. 17). Although *bsk3* deletion alone was sufficient to prevent low N-dependent primary root elongation, strong suppression of N-responsive lateral root elongation required the absence of other closely related BSK3 homologs (Figs. 2 and 5, Supplementary Figs. 6 and 8). These results indicate differential levels of redundancy among BSKs in the regulation of primary and lateral root responses. Redundancy is common for these proteins, as clear growth phenotypes were so far only detected in multiple knockout lines[21]. Our BSK3-GFP translational fusion revealed that BSK3 is present mainly in the epidermis, cortex, and endodermis of the primary root and emerged lateral roots (Fig. 6c, d). Thus, BSK3 localization largely overlaps with that of BR receptors[27,46]. Although we observed N-dependent root growth phenotypes in *bsk3* plants, *BSK3* expression was not regulated by N (Fig. 6 and Supplementary Figs. 15 and 16). This observation suggested that an N-derived signal modulates BR signaling upstream of BSK3. Indeed, *BAK1* was significantly upregulated after prolonged exposure to mild N deficiency (Fig. 7a) and *bak1-1* plants exhibited attenuated root responses to LN (Fig. 7b–d). BAK1 acts on multiple signaling pathways, including a phytosulfokine-dependent signaling cascade that controls root growth[28,47]. However, as overexpression of *BSK3* largely recovered root responses of *bak1-1* plants to LN (Fig. 7e–g), we raised evidence that perturbation of BR signaling is responsible for the defective response of this mutant to mild N deficiency. Recently, it has been shown that prolonged exposure to high temperature promotes primary root elongation in a BRI1-dependent manner[48]. However, in our study, the root phenotypes of two *bri1* mutants indicate that a BRI1-independent mechanism modulates root foraging in response to N (Supplementary Fig. 19). This suggests that at LN other BR receptors, such as BRL1 and BRL3, which share conserved signaling components with BRI1[27], are activated to stimulate BR-dependent root growth.

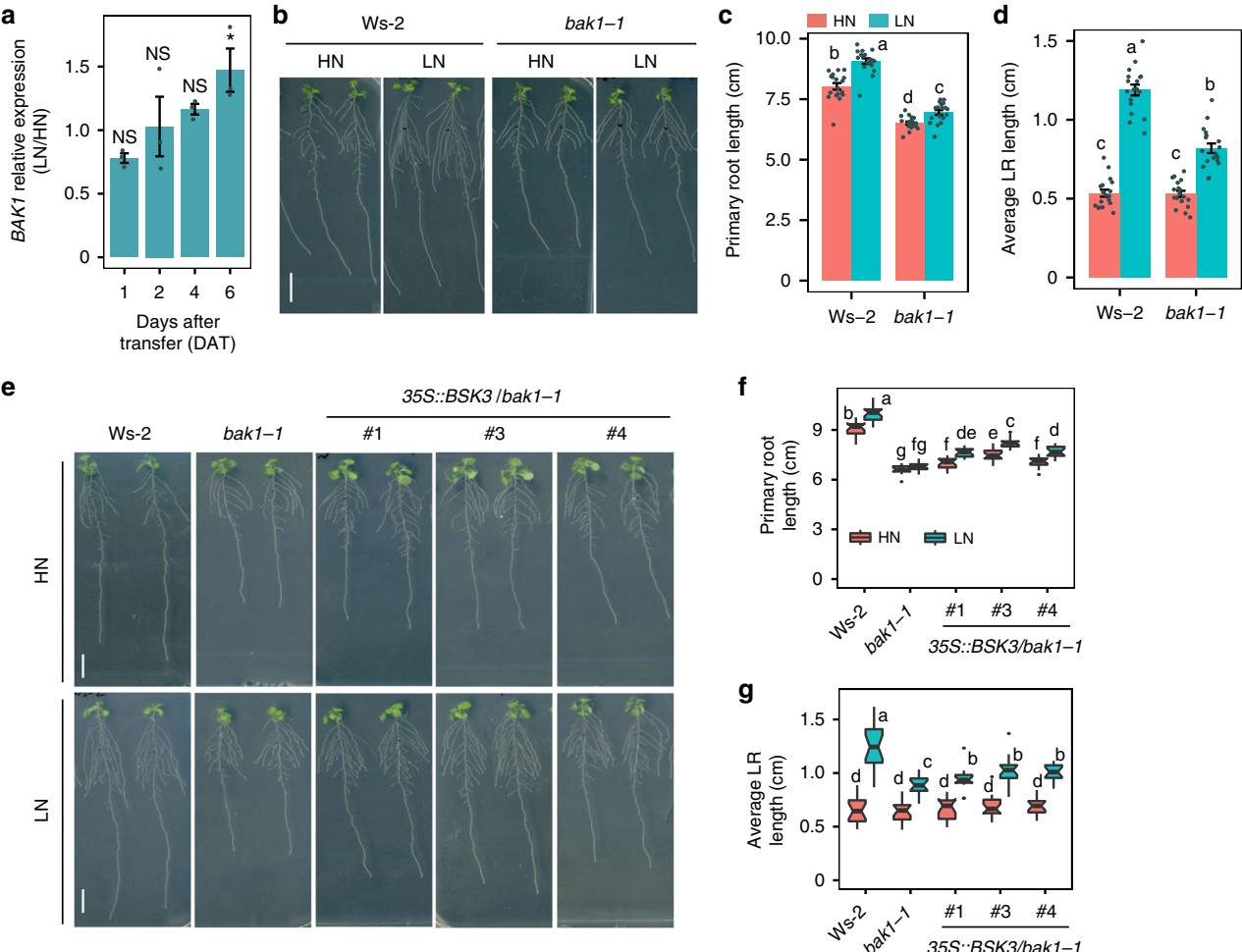

**Fig. 7** Transcriptional regulation of *BAK1* and overexpression of *BSK3* in bak1-1. **a** Fold change of *BAK1* transcript levels in roots during growth on low N. *BAK1* transcript levels were assessed in whole roots by qPCR analysis and normalized to *ACT2* and *UBQ10*. Bars represent means ± SEM (*n* = 3 independent biological replicates). Asterisks indicate significant differences to HN at each time point according to Welch's *t*-test (*$P$ < 0.05; NS, not significant). **b**–**d** Phenotypic analysis of *bak1-1* mutant plants. Appearance of plants (**b**), primary root length (**c**), average lateral root length (**d**) of wild-type (Ws-2) and *bak1-1* plants grown under HN or LN. Bars represent means ± SEM (*n* = 18 independent biological replicates). **e**–**g** Phenotypic analysis of *bak1-1* mutant and BSK3 overexpression lines. Appearance of plants (**e**), primary root length (**f**), and average lateral root length (**g**) of HN- and LN-grown wild type (Ws-2), *bak1-1*, and three independent T2 transgenic lines overexpressing *BSK3* in *bak1-1*. Root system architecture was assessed after 9 days (*n* = 16 independent biological replicates). Horizontal lines show medians; box limits indicate the 25th and 75th percentiles; whiskers extend to 5th and 95th percentiles. Different letters indicate significant differences at *P* < 0.05 according to one-way ANOVA and post-hoc Tukey's test. Scale bars, 1 cm

In agreement with the role of BR signaling in modulating root growth in response to mild N deficiency (Supplementary Fig. 7a–c), we found that L319P substitution in BSK3 is critical for the extent of cell elongation and consequential root-foraging response (Fig. 5a–g). In support of a tight coupling of BSK3-dependent BR signaling and natural variation in root-foraging responses, increased sensitivity to BR conferred by BSK3-L was also accompanied by increased capacity to elongate roots under LN (Fig. 5h, i, Supplementary Fig. 14). Based on these results, we propose that mild N deficiency upregulates the expression of the BR co-receptor gene *BAK1*, thereby activating the BR signaling cascade. Downstream of BAK1, the amplitude of BR signaling is modulated as a result of allelic variation in BSK3, which tunes downstream processes and finally differential root growth through cell elongation. Noteworthy, transcriptome studies have shown that genes involved with BR synthesis, such as *DWARF3/CONSTITUTIVE PHOTOMORPHOGENIC DWARF* (*DWF3/CPD*) and *DWARF4* (*DWF4*), respond to the plant's N status[49–53], suggesting that BR biosynthesis may be also targeted

by N-dependent signaling. In line with this possibility, root elongation under LN was strongly compromised when we blocked BR biosynthesis with brassinazole (Supplementary Fig. 7d, e).

GWA studies not only shed light on the genetic architecture of phenotypic variation but also allow identifying allelic variants that have undergone natural selection due to local stress factors and adaptions beneficial for competitiveness and survival in their natural environments[18,54]. When the topsoil dries out under low precipitation, longer primary roots most likely confer an adaptive advantage as water access from deep soil profiles is facilitated[55]. In fact, also *A. thaliana* plants respond to topsoil drought by developing steeper root systems as a result of enhanced primary root elongation and narrower lateral root angles[56]. Consistent with this, we found that primary root length under low N significantly correlated with maximum precipitation in the wettest month of the natural habitats where the accession lines were sampled (Supplementary Table 2). This raises the question as to whether the natural variation for this trait is, at least in part,

related to water availability. The extensive LD at the BSK3 locus is indicative of a selective sweep (Supplementary Fig. 3), suggesting fixation of beneficial alleles due to natural selection. In accordance with this hypothesis, genome-wide estimation of selection in A. thaliana revealed a strong signature of selection in the BSK3 gene[57]. Interestingly, we found that precipitation in the wettest month, at sites where accessions carrying the BSK3-L-encoding allele were collected, is significantly lower than at those inhabited by accessions harboring the BSK3-P variant (Supplementary Fig. 20). Future research will be necessary to test whether A. thaliana accessions may have adapted to topsoil drought by selection of the stronger BSK3 allele allowing for longer primary roots.

As agricultural plant production must avoid excess N fertilizer input to decrease environmental impacts, there is a quest for crop cultivars that are more efficient in N uptake. The phenotypic variation conferred by the natural allelic variation of BSK3 found in this study could serve as a basis for marker-assisted breeding or allele-specific modifications in crops, allowing roots to explore a larger soil volume as soon as they experience a decrease in their N nutritional status. Exploiting natural allelic variants of BSK3 or generating de novo variants by precise genome editing may thus blaze the trail for developing longer root systems that are better adapted to low N conditions in soils.

## Methods

**Plant materials and growth conditions.** Seeds were surface sterilized in 70% (v/v) ethanol and 0.05% (v/v) Triton X-100, and sown on modified half-strength MS medium (750 μM $MgSO_4 \cdot 7H_2O$, 625 μM $KH_2PO_4$, 1500 μM $CaCl_2 \cdot 2H_2O$, 0.055 μM $CoCl_2 \cdot 6H_2O$, 0.053 μM $CuCl_2 \cdot 2H_2O$, 50 μM KI, 2.5 mM $H_3BO_3$, 2.5 μM MnCl. $4H_2O$, 0.52 μM $Na_2MoO_4 \cdot 2H_2O$, 15 μM $ZnCl_2$) supplemented with 11.4 mM N (1 mM $NH_4NO_3$ + 9.4 mM $KNO_3$), 0.5% (w/v) sucrose, 1% (w/v) Difco agar (Becton Dickinson), and 2.5 mM MES (pH 5.6). Agar plates containing seeds were kept in darkness at 4 °C for 2 days to synchronize germination and then placed vertically in a growth cabinet (Percival Scientific) under a 22 °C/19 °C and 10 h/14 h light/dark regime with light intensity adjusted to 120 μmol photons $m^{-2} s^{-1}$. Homogenous 7-day-old seedlings of representative size for each genotype/independent transgenic line were transferred to new plates with identical sucrose, agar, and nutrient composition as described above, but supplied with either 11.4 mM (HN) or 0.55 mM (LN) N. If not indicated otherwise, plants were cultivated for 9 days on these conditions. All experiments were repeated at least twice. The T-DNA knockout lines SALK_096500C (N666828), SAIL_177_A05 (N862558), SALK_045890C (N659843), SALK_072016C (N671515), SALK_045930C (N657375), SALK_097452C (N673008), SALK_108536C (N653659), SALK_122041C (N658884), SALK_040968C (N662202), SALK_021829C (N682094), SALK_052744C (N678486), SALK_066478C (N698296), SALK_072882C (N675728), SALK_058980C (N678551), eca2-1 (N666473), eca2-2 (N662171), SALK_030563C (N674297), bak1-1 (N6125), and bri1 (SALK_003371C, N678032) were purchased from Nottingham Arabidopsis Stock Center (Nottingham, UK). The bri1-5, bsk3,4, bsk3,4,7, bsk3,4,8, and bsk3,4,7,8 mutants have been described in previous studies[21,58]. The accessions Col-0, Col-3 (N8846), and Ws-2 were used as corresponding wild types for mutant lines as indicated in the figure legends.

**GWA mapping and sequence mining.** For root phenotyping, 200 accessions (Supplementary Data 1) were grown on HN and LN agar plates (4 individual plants per plate). The experiment was repeated 3 times so that a total of 12 plants were analyzed per accession on either N condition. To assess root system architecture, roots were separated until they were clearly distinguishable from one another on the agar and then scanned using an Epson Expression 10000XL scanner (Seiko Epson) with a resolution of 300 dots per inch. Root length was quantified with WinRhizo Pro version 2009c (Regent Instruments). Average values of 12 plants for 200 accessions were calculated and used as phenotypic response in GWAS. We performed GWA mapping using a mixed linear model algorithm implemented in EMMA (Efficient Mixed-Model Association) package as described by Kang et al.[17] and 250 k SNP markers[57,59]. Only SNP with minor allele frequency ≥10% were taken into account. The significant threshold was determined with Benjamini and Hochberg FDR level of $q < 0.1$ for correcting multiple testing[60]. A local association scan with MLMM[19] using stepwise marker inclusion was performed using 236 SNPs spanning a region from 273,948 to 421,221. The BSK3 protein haplogroups were analyzed using 139 genome re-sequenced accessions that were phenotyped in the current experiment. Coding and predicted amino acid sequences were downloaded from A. thaliana 1001 Genomes Project (http://signal.salk.edu/atg1001/3.0/

gebrowser.php). Sequences were aligned with ClustalW2.1 (http://bar.utoronto.ca). Only polymorphisms with minor allele frequency > 5% were considered.

**Transgenic complementation.** For complementation of BSK3, BSK3 promoter (1292 bp from genomic DNA) and CDS of Col-0 (1470 bp from complementary DNA) were amplified using the primers listed in Supplementary Table 3. The amplified fragments were cloned into GreenGate entry modules (pGGA000 for promoter and pGGC000 for open reading frame) and assembled in a pGREEN-IIS-based binary vector following the instructions of Lampropoulos et al.[61]. For allele swapping, the promoter region of BSK3 (2.0 kb upstream of ATG) was amplified from genomic DNA of accessions Col-0 or Cvi-0 and the open reading frames carrying the L or P allele from Col-0 or Co, respectively. All the fragments were cloned separately in GreenGate entry modules and assembled as described above. The Agrobacterium tumefaciens strain GV3101 containing the helper plasmid pSOUP was used to transform plants through the floral dip method[62]. Positive transformants were selected on agar plates supplemented with either 40 mg L$^{-1}$ kanamycin or hygromycin. Following the procedure described earlier for testing BSK-dependent BR responses[20,21], we determined BR sensitivity by measuring the length of primary roots and hypocotyls 6 days after germination. To estimate the primary root growth rate, the root length of the same plants was also assessed after growth for two additional days. To compare the difference between two BSK3 variants, the trait value for each transgenic line was averaged and the difference among means of each variant was compared. There were 14 independent transgenic lines carrying the constructs $proBSK3_{Col-0}::BSK3-L$ or $proBSK3_{Col-0}::BSK3-P$, and 12 for the construct $proBSK3_{Cvi-0}::BSK3-L$ (Figs. 3 and 4a–c, Supplementary Figs. 11 and 12), whereas 17 lines expressed $pro35S:BSK3-L$ or $pro35S:BSK3-P$ in the experiments shown in Fig. 4f–h and 12 lines in the experiment shown in Supplementary Fig. 10. To test the N-dependent root response and N uptake of two BSK3 variants, the growth of six representative lines for each construct was compared with wild type and bsk3,4,7,8 plants. N concentrations in shoot tissues were measured using an elemental analyzer (Euro-EA; HEKAtech).

**Histological analysis and microscopy.** Tissue-specific localization of BSK3 expression was investigated by histological staining of GUS activity in transgenic plants expressing proBSK3-GUS. The GUS-dependent staining pattern was consistent in 12 independent lines. Thus, one representative line was used for detailed analysis. Root samples were stained in 20 mg mL$^{-1}$ (w/v) 5-bromo-4 chloro-3-indolyl-β-D-glucuronic acid, 100 mM $NaPO_4$, 0.5 mM $K_3Fe(CN)_6$, 0.5 mM $K_4Fe(CN)_6$, and 0.1% (v/v) Triton X-100, and incubated at 37 °C for 30–60 min in the dark. Samples were mounted in clearing solution (chloral hydrate:water:glycerol = 8:3:1) for 3 min and imaged using differential interference contrast (DIC) optics on a light microscope (Axio Imager 2, Zeiss).

For root meristem size and cortical cell length measurements, roots were stained with propidium iodide and imaged with a laser-scanning confocal microscope (LSM 780, Zeiss) using 561 nm and 650–710 nm as excitation and emission wavelengths, respectively. For BSK3 protein localization, GFP was excited with a 488 nm Argon laser and emission detected between 505 nm and 530 nm. Quantifications were performed with ZEN software (Zeiss).

**Quantitative real-time PCR.** Root tissues were collected by excision and immediately frozen in liquid N. Total RNA was extracted with RNeasy Plant Mini Kit (Macherey-Nagel GmbH & Co KG, Germany). Quantitative reverse transcriptase PCR reactions were conducted with the CFX38$^{TM}$ Real-Time System and the Go Taq qPCR Master Mix SybrGreen I (Promega) using the primers listed in Supplementary Table 3. Relative expression for all genes was normalized to AtACT2 and AtUBQ10 as internal reference. Relative expression was calculated according to Pfaffl et al.[63].

**Climate data.** We mainly focused on a subset of climate variables gathered by Hancock et al.[64]. Raw data of climate variables (19 climate, latitude and longitude) were downloaded for 115 accessions (Supplementary Data 5) from WorldClim Project (www.Worldclim.org). Pearson's correlation was used to test associations between climate variables and primary root length.

**Statistical analysis.** For correlation analysis, the R function "cor.test ()" was used. Pairwise comparisons between N treatments for gene expression and primary root length were conducted with Welch's t-test. Root traits of different genotypes were compared by one-way analysis of variance (ANOVA) followed by post-hoc Tukey's test at $P < 0.05$. However, for the analyses of transgenic complementation assays, we conducted a Welch's unequal variance t-test on ranked data. This was necessary, because we had different numbers of individuals in each group and because basic assumptions for ANOVA and Tukey's honestly significant difference test were violated due to Mendelian segregation within independent T2 lines and construct-dependent effects in different groups. All statistical analyses were performed in R[65].

**Reporting summary.** Further information on research design is available in the Nature Research Reporting Summary linked to this article.

## Data availability

Data supporting the findings of this study are available within the manuscript and its Supplementary Information files. Datasets generated and analyzed during the current study are available from the corresponding author on reasonable request. The source data underlying Figs. 1–7 and Supplementary Figs. 1–20 are provided as a Source Data file.

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

## Acknowledgements

This work was supported by a scholarship from China Scholarship Council (CSC) to Z.J. We thank Jacqueline Fuge, Annett Bieber, Elis Fraust, and Lisa Gruber (Leibniz Institute of Plant Genetics and Crop Plant Research, Germany) for excellent technical assistance. We further thank Jochen Balbach (University of Halle, Germany) for valuable discussions, as well as Guido Sessa (Tel-Aviv University, Israel) for providing seeds of *bsk3,4*, *bsk3,4,7*, *bsk3,4,8*, and *bsk3,4,7,8* multiple knockout lines and Sunghwa Choe (Seoul National University, South Korea) for *bri1-5* seeds.

## Author contributions

Z.J. performed most experiments. R.F.H.G. helped to prepare plasmids and transgenic plants, supervised qPCR analyses, and imaged roots with confocal microscope. R.C.M. and T.A. provided seeds, SNP data, and information on the location of *Arabidopsis* accessions. Z.J., R.F.H.G., and N.v.W. designed experiments, analyzed the data, and wrote the manuscript. N.v.W. supervised the project.

## Additional information

**Competing interests:** The authors declare no competing interests.

