## [Peer Review File · Nature Communications]

Reviewers' comments:

Reviewer #1 (Remarks to the Author):

This is a very well written manuscript that logically sets out a series of experiments to characterise a BR-related low nitrogen (mild nitrogen stress) foraging mechanism that the authors found to be largely conserved across *Arabidopsis thaliana*. A range of corroborating evidence is well set out by the authors to show that a L319P substitution in the BSK3 gene is causally associated with natural variation of root growth and BR sensitivity.

These findings will certainly be of interest to the root community. It is in line with other studies that show that individual root traits are under individual selection mechanisms, thus advances common hypotheses in the root/development field. It would also be of interest to anyone interested in understanding mechanisms underpinning developmental plasticity to the environment. Sufficient details are given to enable the work to be repeated and the statistical analysis uses appropriate metrics. It was good to see a proper evaluation of the expression location of the BSK3 protein since presence/absence of expression is not the only relevant indicator of a function and more and more we see that redundancy is to do with location of expression level as well as the expression level per se.

Major points:

- 1) One area of information lacking is an understanding of the location of the L/P variation with respect to the 3D structure of the protein. Can the authors carry out some simple bioinformatic analysis to evaluate this? This is very important for understanding why the L/P variation modulates BR sensitivity, i.e. what the crucial underlying mechanism that would mean that one variant enables a more or less sensitive response.
- 2) bsk double/triple mutants show reduced elongation under LN but are also much shorter (significantly for several) than Col-0 or bsk3 (lines 190-197). This effect should be described separately from the lack of LN-foraging-elongation since it likely represents a combination of root growth effects, some of which are N-dependent and others of which are not.
- 3) The finding of a putative correlation of PR length under low N and maximum precipitation, and association between this climatic parameter and the L allele is interesting but requires some testing to link it to the BSK3 effects. For example, analysing BSK3-L variant complementation of bsk3 on different water availability or water supply levels. It would also have been interesting to know whether there were any significant correlated climatic variables when comparing the difference between HN-LN growth (rather than HN or LN individually).
- 4) The authors suggest that BSK1 could be modulated to enable greater crop yield from cultivars with larger root systems, but they do not present any evidence to show that the longer root actually enables more N uptake. N measurements in the shoots of these seedlings would be needed for this argument to be substantiated. In the context of crop improvement it would also be useful to know if the longer primary root production negatively impacts another aspect of growth (due to more C/energy/tradeoff needed for larger growth); however this could be in future work and is not needed for this publication.

Minor points:

Some phenotyping is carried out at 6/8 days and some at 9 days (with or without pre-transfer at 7 days). Is there a reason for this variability? I don't think it necessarily needs to be consistent, but clarification is needed.

Line 42: Please re-write this sentence to show what evidence is provided by which reference in the long reference list (5-7,11,12).

Section from line 122: Although Fig.1 seems to show just one significant SNP hit that is then followed up on, it would be very helpful to provide an overview of the SNP landscape. At lower significance cutoff (which is sensibly chosen here) there are several more peaks that could be informative to know about in future work (e.g. in a larger GWAS study).

Line 182: Remove 'most importantly' or change to 'To confirm that loss of BSK3 expression was responsible for the phenotype seen'.

Line 405: What does 'at the population scale' mean here? Just because the allelic variation is seen over a population it doesn't mean that the response is at the population scale. Perhaps rephrasing would help.

Line 452: You mention 'in a previous study' but give two references: please clarify.

Line 500: I'm not sure if the 'degree of freedom' is a useful way of describing this – perhaps just say there were 14 or 17 transgenic lines for each.

Section from lines 557 (References): Just a few formatting amendments to make.

Figure 8: More description of the panels in c would be useful since there are variations in BSK3 expression over time and in different cell types. There seems to be higher BSK3 expression at the emerged lateral root stage (middle panel) at LN, but perhaps this was not significant?

Figure S1: Can you provide a higher-resolution and larger map image here to more easily see the location of accessions over the countries?

Figure S3: Which allelic version do each of these accessions have? This figure does not seem to be referred to in the text and it is not clear what gene expression data/analysis this refers to.

Figure S6: Please add the names of the climate variables into the table (i.e. bio13 = precipitation during the wettest month).

Figure S7: Please clearly define what 'progressing N deficiency' is - - N deficiency in the plant or the environment? Do you know that one or both of these are progressing (i.e. becoming more N deficient over time), using [N] measurements? For ease of reading please define DAT in the figure legend.

It would be useful to provide a graphical summary of the molecular interactions described in lines 350-358/390-395 related to the effects of low N.

Reviewer #2 (Remarks to the Author):

In the manuscript "Natural variation of BSK3 tunes brassinosteroid signaling to regulate root foraging responses to low nitrogen" Zhongtao Jia and colleagues uncover the role of Brassinosteroid signaling and in particular the gene BSK3 for low-Nitrate induced root growth foraging patterns. They further provide evidence that natural variants of BSK3 are causing natural variation of root growth responses to low N, and moreover find evidence that selective pressures that are correlated with precipitation parameters might favor one allele over the other. Overall, this is an impressive and important work that uncovers new and exciting biology, as well as provides potential for meaningful agricultural application as improving nitrate related root foraging traits might improve crop yields and reduce inputs and environmental damage.

The manuscript is written very well and in a concise manner. The research is of high quality and the scope of the work is impressive. Most of the conclusions are supported by the data. I truly enjoyed reviewing this manuscript.

I only have few concerns and suggestions.

Major:

1. My main concern relates to the link of BSK3 to the GWAS data. The GWAS peak (derived from the 250K SNP dataset) is detected approx. 93KB downstream of BSK3. To find BSK3, the authors took into account a huge region using a very low LD threshold and then filter out the list of candidate genes according to expression and mutant phenotypes. After they found positive evidence for BSK3, they also look at LD using the 1001 genomes data and find a SNP that leads to a BSK3 mutation that is interpreted as causal. While this BSK3 SNP is highly linked to the GWAS top SNP, it is not significantly associated in a GWAS conducted on the 1001 genomes data. The authors also perform some kind of haplotype analysis to strengthen their case, which is never shown nor described. Without any further experimentation, the involvement of BSK3 variants would be questionable but because of the very commendable allelic complementation analysis, it can be concluded that BSK3 allelic variation is contributing to the observed natural variation. However, I don't think this clearly shows whether the BSK3 variants are underlying the GWAS peak. I think the authors will need to clarify this.

i) Is the GWAS peak perhaps a ghost peak that is caused by multiple alleles? The Nordborg lab has recently published a nice case (Kerdaffrec et al 2016 eLife). Perhaps one could see if a multi-locus GWAS based on stepwise inclusion of markers could expose a similar phenomenon in this case (such as procedure is possible on the GWAPP portal). Also, a figure with the haplotypes in this region might be helpful to illuminate this issue.

ii) Perhaps the single trait GWAS was confounding? Would a multi-trait GWAS be more appropriate or a ratio of HN/LN traits for the analysis and expose the BSK3 SNP?

iii) Might population structure correction interfere with the significance of the BSK3 SNP? Perhaps the authors could consider running a simple linear model GWAS to check?

iv) Can the authors really rule out that there is not a gene closer to the SNP and therefore more in line with the expected LD breakdown in Arabidopsis that is responsible for the GWAS peak? It could be a gene that is not contained in the RootMap (several thousand of genes are missing as it was ATH1 microarray based) or expressed below the detection level or expressed only transiently (diurnal or upon certain N conditions).

v) In any case, the issues related to the large distance between top SNP and BSK3, as well as potential other explanations, should be discussed in the discussion and not be swept under the rug. I don't think discussing alternatives diminishes the story – it will rather increase its merit.

2. While being very interesting, the link to environmental parameters is not worked out well. First of all, it is not clear how the correlation analysis was conducted. In line 272 the authors state they relate haplotypes to the climate parameters. However, there is no hint how these haplotypes were computed, nor what they really are. Also, how does the GWAS top SNP and the BSK3 SNP relate to this analysis. Second, "This suggests that BSK3 variants were potentially selected by precipitation." is worded very imprecisely. How can precipitation select something? Third, in the discussion the authors relate precipitation to drought, but precipitation can as well affect the leaching of nitrate by strong rainfalls or denitrification plus a huge array of other biotic and abiotic factors. This should be discussed. Fourth, the method regarding the correlation is unclear. Please add detail on the linear model and what was done.

Minor:

1. It would be nice to check the BR sensitivity of contrasting accessions for N-root responses and correlate this with the response to N; this could strengthen the case for causality of BSK3 further.

2. Line 309: The authors should outline the nature of the *bri1* mutants (is it loss of function?). Also, how can the authors reconcile the lack of phenotype in the *bri1* mutants with the function of BSK3 and BAK1 in the N-dependent response? That should be discussed.
3. Line 322 (and perhaps throughout the text): BSK3 is not causal for the observed variation. It is most likely causal for part of the observed variation. Also, it would be nice if the authors could estimate somewhere how much of the phenotype the GWAS SNP or the BSK3 SNP explains.
4. Line 332: change allelic variation to allelic variant. Also, are the authors sure there is only one variant?

Reviewer #3 (Remarks to the Author):

Major advances:

- GWA identifies BSK3 and a non-synonymous SNP to be associated with variation in primary root length response in Arabidopsis accessions
- the first time that BR has been shown to be a component of regulation of root system architecture in response to differing levels of available N
- elegant analysis showing that this precise allelic variation contributes to variation in a variety of root system architecture traits, relative to Col-0

Major criticisms:

- the majority of data supporting this particular amino acid change in BSK3 as being linked to variation in the LN, is via tests in Col-0. However, Col-0 is the accession that shows a mid-range trait, and the mutant phenotype of the GSK3 is more similar to the accessions which do NOT show a LN response for primary root length (Petergof, Tsu-0 and Nd-1). Thus, I do not believe that the data provided supports GSK3 as being broadly causal for the variation observed in the N-mediated changes in root system architecture. Further, do Petergof, Tsu-0 and Nd-1 contain the P or the L haplotype in GSK3? Why was the genomic complementation not done with the opposing allele in one of the most extreme accessions?
- the sudden jump to exploring lateral root elongation and lateral root density is not supported by the original GWA screen. I suggest that the authors also refer to Gifford et al., 2013 in this respect. Furthermore, with respect to the lateral root traits, differences in the magnitude of a response are being measured. Thus, pairwise statistical comparisons are not suitable and instead an ANOVA should be carried out looking at genotype x treatment interactions, followed by post-hoc Tukey tests. This should be done for all experiments involving lateral root traits.
- the fact that the P/L haplotypes regulate root length under all N treatments is also quite confusing. I appreciate that it does indicate that there is allelic variation, however, the link back to the original accessions that showed the most extreme responses is not made.
- the link with the precipitation is not strong and does not add to your story. There is no evidence in the literature for excessive wetness or flooding to be related to LN – this should either be tested or removed

Minor criticisms:

- a comparison of their associations relative to those in the Gifford et al. study would provide context for their identified SNPs and novelty
- it should be stated in the main text what the N source is (NO₃ + NH₄)
- link between the accessions with no response, to the accessions with the greatest response, and the majority
- in lines 111 and 112 – it should be clarified that this was only with a subset of the accessions and cannot necessarily be extrapolated to the whole population
- is there any evidence for BR signaling as being regulated by N in the many transcriptional meta-analyses performed by G. Krouk and R. Gutierrez?

- line 132 – this could be stated more clearly – that a gene needs to at least be expressed in one tissue in the root
- is there any GWA signal for the other BSK genes in your study or in the Gifford study?
- lines 205 to 206 are not supported by the data

Reviewer #4 (Remarks to the Author):

In this manuscript, Jia et al. showed that BSK3 regulates root foraging response under mild nitrogen deficiency condition. Authors analyzed primary root growth against nitrogen deficiency in 200 accessions of Arabidopsis and identified a genetic locus (BSK3) responsible for root foraging response from GWAS analysis. Authors insist that allelic variation (L319P) on BSK3 protein alters root elongation and BR sensitivity. Although authors tried to find genetic loci regulating root foraging response with interesting screening approach, the current manuscript has serious problems and many issues to be further addressed.

<Manuscript organization and reliability>

1. Surprisingly, some images of the results are used in duplicate more than once. For example, phenotype images of *bsk3,4,7,8* in Figure 5a and Figure S4b are the same (but the scale is different). A more serious problem is that the authors used the same measurement in different experiments. Primary root length of *bsk3,4,7,8* (n=123) shown in Figure 5a are exactly the same as that of Figure S4c (*bsk3,4,7,8*, n=123) although they are independent experiments. In different experiments, authors seem to use the same control value. This is a serious problem which considered as a scientific misconduct.

Followings are other cases which I found in the manuscript.

- An image of #9 *proBSK3::BSK3-L/bsk3,4,7,8* line (Figure 5a) is found in Figure 6a.
- Two images (*Col-0* and *bsk3,4,7,8* treated by BL) shown in Figure 6a is duplicated in Figure S5a. In addition, the same values of primary root length were used in both Figure 5b and Figure S5b.
- #6 (*proBSK3::BSK3-L/bsk3,4,7,8*) line of Figure S4b is duplicated in Figure 6a.

2. Although authors used multiple transgenic lines (usually three independent lines) and showed their phenotypes, in measurement, authors seem to combine the values all of the lines and present their mean as a single transgenic line. Unlike controls, error bar is too big although n value is very high. It is doubtful that three independent line showed a different pattern for root growth.

3. In all experiments using transgenic plants expressing BSK3, authors should compare BSK3 protein level. Regardless of the native promoter or 35S-promoter, gene expression is affected by the insertion position and copy number of T-DNA on the chromosome. Authors can compare BSK3-L and BSK3-P activity based on the similar protein level.

4. Authors investigated root foraging responses of 200 accession of Arabidopsis under low N condition (Supplemental Table 1), selected 9 ecotypes, and measured meristem size, cortical cell length. Why did the authors show only 3 ecotypes' meristem images although authors showed all 9 ecotypes' cortical cell length? In addition, authors need to show LN response of 9 ecotypes in main figure (prior to Figure 1b).

5. From Figure 1, authors divided 9 accessions into three categories based on their root foraging response to low N condition. However, their responses are not tightly correlated with a.a. 319 variation of BSK3. Weak/no: *Petergof* (P), *Tsu-0* (P), *Nd-1* (L); intermediate: *Co*(P) and *Col-0* (L); strong: *LDV-14* (NA), *LDV-58*(NA), *Do-0*(P) and *Cvi-0*(P). How can authors explain this contradiction? Later, authors' experiments were just focused on *Col-0* carrying BSK3-L but mild response to low N. Authors compared promoter activity between *Col-0* and *Cvi-0*. However, *Cvi-0* showed strong LN-induced root elongation although it contains BSK3-P. It is difficult to understand the logic to design the overall experiment.

<This reviewer could not find any concrete evidence that BR regulates LN-induced root growth>

1. In Figure 2e, although primary root length of L type BSK3 was slightly longer than P type BSK3

in HN as well as LN condition, the ratio of HN/LN between L type and P type look almost similar (regardless of the amino acid type, root elongation ratio in response to low N is similar). This means that variation of BSK3 sequence is not major reason for different foraging response.

2. Authors described that cortical cell size correlates with strong response for LN condition. It is known that BR promotes root elongation in epidermal layer. Is there any specific function of BR in cortical cells of root? Authors need to explain the BR action in cortical cells.

3. Figure 6d and e, authors need to test BR sensitivity for 35S-BSK3-P or 35S-BSK3-L.

4. Based on the results of complementation experiments, authors argued that BSK3-L is more functional in BR signaling and root foraging response than BSK3-P. What is the mechanism underlying different response? This variation affects on interaction of BSK3 with known BR signaling component or kinase activity?

5. In Figure 5 and 6, current results just partly support that BSK3-L form is more functional in BR signaling. Have authors tested BL or Brassinazole effect on LN-induced root elongation?

6. Authors suggested that low N activates BR signaling through elevation of BAK1 mRNA level. BAK1 is multifunctional co-receptor which interacts with several RLKs. the phenotype of bak1 mutant which shows a subtle difference (Figure 9) can't support that BR regulates N deficiency response of roots. In addition, BSK members also are regulated by another RLKs such as FLS2. Most of all, Figure S8 indicates that bri1 does not show any difference in LN-induced root growth. BSKs are activated by BRI1 but not by BAK1. If the phenotype of the bsk3 mutant was due to the defect of BR signaling, the bri1 mutant must show the same phenotype.

Reviewer #1 (Remarks to the Author):

This is a very well written manuscript that logically sets out a series of experiments to characterise a BR-related low nitrogen (mild nitrogen stress) foraging mechanism that the authors found to be largely conserved across *Arabidopsis thaliana*. A range of corroborating evidence is well set out by the authors to show that a L319P substitution in the BSK3 gene is causally associated with natural variation of root growth and BR sensitivity.

These findings will certainly be of interest to the root community. It is in line with other studies that show that individual root traits are under individual selection mechanisms, thus advances common hypotheses in the root/development field. It would also be of interest to anyone interested in understanding mechanisms underpinning developmental plasticity to the environment. Sufficient details are given to enable the work to be repeated and the statistical analysis uses appropriate metrics. It was good to see a proper evaluation of the expression location of the BSK3 protein since presence/absence of expression is not the only relevant indicator of a function and more and more we see that redundancy is to do with location of expression level as well as the expression level per se.

Response: Thanks for the encouraging comments about our manuscript. To investigate BSK3 protein localization, we generated a *proBSK3::BSK3-GFP* translational fusion construct, which was introduced in Col-0 plants. Confocal analysis of independent T2 lines revealed that the protein was present from the meristematic zone all the way through the elongation zone and was primarily localized in epidermal, cortical and endodermal cells, most likely in the plasma membrane. These data are presented in new Fig. 6c-d and described in the text, in lines 291-295: "Confocal microscopy of plants expressing *proBSK3::BSK3-GFP* further revealed that the protein was present from the meristematic zone all the way through the elongation zone of epidermal, cortical and endodermal cells (Fig. 6c). Confined localization at the cell border was in agreement with the recently confirmed binding of the myristoylated protein to the plasma membrane³¹."; and lines 301-303: "Under both N conditions, BSK3 promoter activity and BSK3-GFP fusion protein were only detected in lateral roots that had emerged from the parental root (Fig. 6d; Supplementary Fig. 14)."

Major points:

1) One area of information lacking is an understanding of the location of the L/P variation with respect to the 3D structure of the protein. Can the authors carry out some simple bioinformatic analysis to evaluate this? This is very important for z why the L/P variation modulates BR sensitivity, i.e. what the crucial underlying mechanism that would mean that one variant enables a more or less sensitive response.

Response: Thanks for this suggestion. We now tried to model our BSK3 variants together with Prof. Dr. Jochen Balbach (Martin-Luther-Universität Halle-Wittenberg) based on the existing partial crystal structure of BSK8 (residues 40-328), which mainly contained the predicted kinase domain of the protein (Grütter et al., 2013, J Mol Biol). By homology modeling, we could overlay the BSK3 protein (in cyan) on the BSK8 structure (in green) (Fig. 1a to reviewers). We then compared the structures of the two BSK3 variants and found that there were no substantial differences in the local structure (Fig. 1b to reviewers). However, our model indicates that the amino acid residue 319 (L or P) is located on the surface of the protein (Fig. 1c to reviewers). In addition, it appears that this residue is located in a loop just in front of a short α -helix. Unfortunately, since the partial structure of BSK8 ends a few

amino acids downstream of position 319, it remains unclear if BSK3's residue 319 is indeed located on a loop or if the short α -helix actually extends even further than predicted. Having exactly this information is very important, as a leucine to proline substitution within an α -helix would have a significant consequence for the protein conformation because proline typically breaks the α -helical conformation. Thus, considering that our modeling was based on a partial structure, we feel that we cannot draw any firm conclusion from this analysis and prefer not to integrate these data in the revised manuscript.

Figure 1 to reviewers. Homology modeling of BSK3 protein variants to the existing partial crystal structure of BSK8. (a) Structural overlay of BSK3-L (in cyan) with the available partial structure of BSK8 (in green). **(b)** Comparison of local structure of BSK3-P (in magenta) and BSK3-L (in cyan) based on homology to BSK8. **(c)** A zoom-in view of the region highlighted in the red frame in Fig. 1b. A white arrow indicates the location of the 319 L/P substitution.

2) *bsk* double/triple mutants show reduced elongation under LN but are also much shorter (significantly for several) than Col-0 or *bsk3* (lines 190-197). This effect should be described separately from the lack of LN-foraging-elongation since it likely represents a combination of root growth effects, some of which are N-dependent and others of which are not.

Response: Yes, we agree with this point. We now tried to make this point clearer in the revised text, as follows (lines 182-194): “In line with the partially redundant function of these most closely related BSKs³⁰, the *bsk3,4,7,8* quadruple mutant exhibited the strongest decrease in root length as compared to the *bsk3* single mutant at both N conditions (Fig. 2a-c). Only minor effects were observed for *bsk3,4* double and *bsk3,4,7* triple mutants at HN condition, while no effect was observed for *bsk3,4* double, and *bsk3,4,7* or *bsk3,4,8* triple mutants at LN. Notably, the primary root length in response to LN was similarly attenuated in all tested mutants, except for *bsk3,4,7* (Fig. 2a, b). The additive effect of all four BSKs was particularly relevant for average lateral root length, as *bsk3,4,7,8* plants failed to significantly elongate lateral roots at LN (Fig. 2c). Consequently, these plants showed no significant increase in total root length under LN supply (Supplementary Fig. 7a). Relative to Col-0, none of the tested mutants showed significant changes in lateral root density (Supplementary Fig. 7b).”

3) The finding of a putative correlation of PR length under low N and maximum precipitation, and association between this climatic parameter and the L allele is interesting but requires some testing to link it to the BSK3 effects. For example, analysing BSK3-L variant complementation of *bsk3* on different water availability or water supply levels. It would also have been interesting to know whether there were any significant correlated climatic variables when comparing the difference between HN-LN growth (rather than HN or LN individually).

Response: For the present study we were unable to further explore this aspect, as we first need to establish an experimental growth system that can simulate differential water availability along a vertical soil profile. Considering the comments of Reviewers 2 and 3, we decided to point to this observation only in the Discussion (lines 453-466), as this provides an interesting starting point for future research. The data are now shown in Supplementary Fig. 18. As suggested by the reviewer, we assessed the correlation between primary root response to low N (LN-HN) and climate variables but found no significant correlations.

4) The authors suggest that BSK1 could be modulated to enable greater crop yield from cultivars with larger root systems, but they do not present any evidence to show that the longer root actually enables more N uptake. N measurements in the shoots of these seedlings would be needed for this argument to be substantiated.

Response: We assessed the shoot N concentrations of Col-0, *bsk3,4,7,8* mutant and six transgenic lines expressing the BSK3-P and six expressing the BSK3-L variant in the *bsk3,4,7,8* quadruple mutant (Fig. 2 for reviewers). At low N supply, shoot N levels were indeed significantly lower in the *bsk3,4,7,8* quadruple mutant as compared to the wild type. However, the expression of either BSK3 variant in the quadruple mutant did not significantly increase shoot N concentration, despite differentially altering root architecture of plants (e.g., Fig. 5a-d in the manuscript). This result is not surprising, as we expect that the benefits of a larger root system for N capture will become more apparent in soil conditions, where the vertical movement of N and water along the soil profile imposes a challenge to plants that we cannot easily simulate in our agar plate system. Nevertheless, we base our proposition in the Discussion on several studies showing a positive correlation between a large root system and increased N uptake in crop plants (Mu et al., 2015, Eur J Agron; Gamuyao et al., 2012, Nature; Ma et al., 2017, Plant Physiol).

Figure 2 for reviewers. Shoot N concentrations of plants grown under different N availabilities.

Seven-day-old seedlings were pre-cultured on 11.4 mM N and then transferred to solid agar containing either high N (HN, 11.4 mM N) or low N (LN, 0.55 mM N). Shoots were collected for N concentration determination 9 days after transfer. Bars represent means \pm s.e. (n = 6 biological replicates for Col-0 and *bsk3,4,7,8*, and 3 replicates for individual transgenic allelic complementation lines). Different letters indicate significant differences at $P < 0.05$ according to one-way ANOVA and post-hoc Tukey test.

In the context of crop improvement it would also be useful to know if the longer primary root production negatively impacts another aspect of growth (due to more C/energy/tradeoff needed for larger growth); however this could be in future work and is not needed for this publication.

Response: We agree with the reviewer that this aspect deserves consideration in a future study. There are encouraging examples in the literature showing that root growth can be stimulated without negatively affecting grain yield (e.g., Gamuyao et al., 2012, *Nature*; Ma et al., 2017, *Plant Physiol*; Ramireddy et al., 2018, *Plant Physiol*). In fact, it has been shown that the increased metabolic cost of producing longer roots can, at least in part, be compensated for by altering other traits/processes, such as decreased root respiration and/or root thickness (reviewed by Lynch, 2015, *Plant Cell Environ*).

Minor points:

Some phenotyping is carried out at 6/8 days and some at 9 days (with or without pre-transfer at 7 days). Is there a reason for this variability? I don't think it necessarily needs to be consistent, but clarification is needed.

Response: We apologize for not clearly explaining these differences in the first version. In order to avoid possible confounding effects arising from putative differences in seed N concentrations, in experiments assessing the effect of N supply, we always pre-cultured our plants on sufficient N for 7 days before transferring them to N treatments. Since the root foraging response to LN appears after more prolonged cultivation, we harvested plants in all N experiments 9 days after transferring them to HN or LN. In experiments with exogenous supply of 24 epi-brassinolide (BL), in which we were mainly interested in the short-term response of plants to BRs and did not test different N conditions, we germinated plants directly on -BL or +BL. In this particular experiment, we assessed root traits of the same plants 6 and 8 days after germination, in order to calculate the root elongation rates. We have now clarified these differences in the revised Methods (lines 484-488): "Seven-day-old seedlings of similar size were transferred to new plates with identical sucrose, agar and nutrient composition as described above but supplied with either 11.4 mM (HN) or 0.55 mM (LN) N. If not indicated otherwise, plants were cultivated for 9 days on these conditions." and (lines 536-541): "For BR sensitivity assays, seeds were directly sown on solid agar plates supplemented with 1 μ M 24-epibrassinolide (BL; CAS 78821-43-9, Sigma) dissolved in pure ethanol. Mock treatments received the same amount of solvent. The length of primary roots and hypocotyls were measured 6 days after germination. To estimate the primary root growth rate, the root length of the same plants was also assessed after growth for two additional days."

Line 42: Please re-write this sentence to show what evidence is provided by which reference in the long reference list (5-7,11,12).

Response: Thanks for pointing this out. The sentence now reads (lines 40-42): "Whereas nitrate (NO_3^-) mainly stimulates lateral root elongation^{5,6,11,12}, ammonium (NH_4^+) induces lateral root branching⁷, supporting the view that these two major inorganic N forms shape root system architecture in a complementary manner."

Section from line 122: Although Fig.1 seems to show just one significant SNP hit that is then followed up on, it would be very helpful to provide an overview of the SNP landscape. At lower significance cutoff (which is sensibly chosen here) there are several more peaks that could be informative to know about in future work (e.g. in a larger GWAS study).

Response: Thanks for this suggestion. We now show SNPs with $-\log P$ -value >4 in the new Supplementary Table 2. In total, 53 SNPs were found to be significant at this significance cutoff. Of these, 24 were in the region of the major peak associated with primary root length at low N. In

addition, we detected a SNP on chromosome 3 located in *PHOSPHATE1 (PHO1)*, a gene previously shown to modulate root allometry in response to nitrate and hormones (Rosas et al., 2013, PNAS). We then grew the *pho1-2* mutant under our conditions and observed that the mutant's primary root also fails to respond to mild N deficiency. These new results were integrated in Supplementary Fig. 2 and described in the manuscript (lines 111-116) "The associated SNP on the chromosome 3, which accounted for 6.3% of the natural variation for primary root length at LN, was located in *PHOSPHATE 1 (PHO1)*, a gene previously shown to modulate root allometry in response to auxin, ABA and nitrate²⁷. The *pho1-2* mutant exhibited a strong reduction of primary root response to LN (Supplementary Fig. 2), supporting the hypothesis that *PHO1* was the underlying gene for this locus."

Line 182: Remove 'most importantly' or change to 'To confirm that loss of BSK3 expression was responsible for the phenotype seen'.

Response: Thanks. In order to accommodate all new data in the manuscript, we have now moved this dataset to Supplementary Fig. 6 and shortened the sentence to (lines 173-176): "Notably, the elongation of both primary root and lateral roots in response to LN could be largely rescued by introducing a genomic fragment containing the promoter and coding regions of *BSK3* into the *bsk3* mutant (Supplementary Fig. 6)."

Line 405: What does 'at the population scale' mean here? Just because the allelic variation is seen over a population it doesn't mean that the response is at the population scale. Perhaps rephrasing would help.

Response: We agree. The sentence now reads (lines 439-441): "Downstream of BAK1, the amplitude of BR signaling is modulated as a result of allelic variation in BSK3, which tunes downstream processes and finally the differential root foraging response through cell elongation."

Line 452: You mention 'in a previous study' but give two references: please clarify.

Response: Thanks for spotting this. We have now corrected the sentence as follows (lines 496-497): "The *bri1-5*, *bsk3,4*, *bsk3,4,7*, *bsk3,4,8* and *bsk3,4,7,8* mutants have been described in previous studies^{30,61}."

Line 500: I'm not sure if the 'degree of freedom' is a useful way of describing this – perhaps just say there were 14 or 17 transgenic lines for each.

Response: As suggested, we revised the sentence to (lines 543-548): "There were 14 independent transgenic lines carrying the constructs *proBSK3_{Col-0}::BSK3-L* or *proBSK3_{Col-0}::BSK3-P*, and 12 for the construct *proBSK3_{Cvi-0}::BSK3-L* (Fig. 3 and 4a-c; Supplementary Figs. 11 and 12), whereas 17 lines expressed *pro35S::BSK3-L* or *pro35S::BSK3-P* in the experiments shown in Fig. 4f-g, and 12 lines in the experiment shown in Supplementary Fig. 10."

Section from lines 557 (References): Just a few formatting amendments to make.

Response: Thanks. Done!

Figure 8: More description of the panels in c would be useful since there are variations in BSK3 expression over time and in different cell types. There seems to be higher BSK3 expression at the emerged lateral root stage (middle panel) at LN, but perhaps this was not significant?

Response: We checked this possibility with the newly generated *proBSK3::BSK3-GFP* lines (new Fig. 6d), but found no N-dependent effect on *BSK3* expression pattern or BSK3-GFP protein fluorescence in dependence of the developmental stage.

Figure S1: Can you provide a higher-resolution and larger map image here to more easily see the location of accessions over the countries?

Response: Yes. Since most of the accessions used in this study originate from Europe, in the new Supplementary Fig. 1 we included a second panel with a zoomed-in view of this region. The approximate location and the countries in which each accession was sampled are now indicated by color-coded dots. Moreover, we have also included information about the collection site (latitude, longitude and country) in Supplementary Table 1.

Figure S3: Which allelic version do each of these accessions have? This figure does not seem to be referred to in the text and it is not clear what gene expression data/analysis this refers to.

Response: We refer to this figure (now Supplementary Fig. 5) in lines 140-142: "To further ascertain the causal gene for the associated locus, we assessed the expression of *ASKO* and *BSK3* in 9 accessions that showed contrasting primary root lengths (Supplementary Fig. 5)."

In the revised Supplementary Fig. 5, we also indicate the allelic versions of *BSK3* in the selected accessions (i.e. Kas-2, Co, Ty-0, Tamm-27, JEA and Uod-1 express the P-type protein, while Edi-0, Ven-1 and Col-0 express the L-type).

Figure S6: Please add the names of the climate variables into the table (i.e. bio13 = precipitation during the wettest month).

Response: Done! This figure is now Supplementary Fig. 18.

Figure S7: Please clearly define what 'progressing N deficiency' is - - N deficiency in the plant or the environment? Do you know that one or both of these are progressing (i.e. becoming more N deficient over time), using [N] measurements? For ease of reading please define DAT in the figure legend.

Response: In order to refer more clearly the conditions tested in the experiment, we now replaced "progressing N deficiency" in the text by "prolonged N deficiency (line 298)". In order to integrate the new data, we also changed the title of Fig. 7 to: "Transcriptional response of *BAK1* to low N and overexpression of *BSK3* in the *bak1-1* mutant."; of Supplementary Fig. 13 (former Supplementary Fig. 6) to: "Time course of *BSK3* expression in response to low N."; and Supplementary Fig. 17 (former Supplementary Fig. 7) to: "Transcriptional response of *BR11* to low N and phenotypic analysis of *bri1* mutant plants." In addition, we indicated in all figure legends that "DAT" refers to "days after transfer".

It would be useful to provide a graphical summary of the molecular interactions described in lines 350-358/390-395 related to the effects of low N.

Response: Depicting molecular interactions emphasizing the role of the P/L substitution in *BSK3* is currently very difficult, as long as we don't know the precise biochemical function, in which the identified substitution is involved. Moreover, it is also difficult to depict an enhanced sensitivity in the BR signaling cascade, as it is mediated by the *BSK3*-L-variant, as long as we don't know how signaling output by *BSK3* is modulated in a quantitative manner. We therefore resign from presenting a graphical sketch.

Reviewer #2 (Remarks to the Author):

In the manuscript “Natural variation of BSK3 tunes brassinosteroid signaling to regulate root foraging responses to low nitrogen” Zhongtao Jia and colleagues uncover the role of Brassinosteroid signaling and in particular the gene BSK3 for low-Nitrate induced root growth foraging patterns. They further provide evidence that natural variants of BSK3 are causing natural variation of root growth responses to low N, and moreover find evidence that selective pressures that are correlated with precipitation parameters might favor one allele over the other. Overall, this is an impressive and important work that uncovers new and exciting biology, as well as provides potential for meaningful agricultural application as improving nitrate related root foraging traits might improve crop yields and reduce inputs and environmental damage.

The manuscript is written very well and in a concise manner. The research is of high quality and the scope of the work is impressive. Most of the conclusions are supported by the data. I truly enjoyed reviewing this manuscript.

I only have few concerns and suggestions.

Response: Thanks for the encouraging comments about our work.

Major:

1. My main concern relates to the link of BSK3 to the GWAS data. The GWAS peak (derived from the 250K SNP dataset) is detected approx. 93KB downstream of BSK3. To find BSK3, the authors took into account a huge region using a very low LD threshold and then filter out the list of candidate genes according to expression and mutant phenotypes. After they found positive evidence for BSK3, they also look at LD using the 1001 genomes data and find a SNP that leads to a BSK3 mutation that is interpreted as causal. While this BSK3 SNP is highly linked to the GWAS top SNP, it is not significantly associated in a GWAS conducted on the 1001 genomes data. The authors also perform some kind of haplotype analysis to strengthen their case, which is never shown nor described. Without any further experimentation, the involvement of BSK3 variants would be questionable but because of the very commendable allelic complementation analysis, it can be concluded that BSK3 allelic variation is contributing to the observed natural variation. However, I don't think this clearly shows whether the BSK3 variants are underlying the GWAS peak. I think the authors will need to clarify this.

Response: Thank you for these suggestions. We agree that the LD threshold (\$r^2 > 0.2\$ ) used in our GWAS is on the lower side. However, we chose this LD threshold, because the average whole genome LD in *Arabidopsis thaliana* has been estimated to 10 kb at \$r^2 > 0.2\$ (Kim et al., 2007, Nat Genet). We nevertheless followed the reviewer's recommendation and re-analyzed the data using a higher LD threshold (\$r^2 > 0.7\$ ). Also with this threshold BSK3 was still contained within the obtained 105 kb-long linkage block. Noteworthy, although C956T was not included in the 250 k GWA SNP array, the G1353A polymorphism in BSK3's coding sequence (which leads to a synonymous substitution, Fig. 1h) was present in the array and showed a moderate association with primary root length (\$-\log_{10}(\text{p-value}) = 4.1\$ ). In addition, G1353A also showed strong LD (\$r^2 > 0.7\$ ) with our top GWA peak. This was also the reason why we took BSK3 as candidate for T-DNA mutant characterization. It is not uncommon that the causal gene resides outside the expected LD window (i.e. 20 kb centered to the peak SNP) as shown in the GWA studies of Chao et al. (2014, PLoS Biol), Bac-Molenaar et al.

(2015, Plant Cell) and van Rooijen et al. (2017, Nat Commun). One explanation is that the LD varies significantly especially within genomic regions undergoing natural selection. In line with this, we found a long LD range in the *BSK3* locus (Supplementary Fig. 3). Similar to our case, van Rooijen et al. (2017, Nat Commun) identified the causal gene *YELLOW SEEDLING 1*, which was associated with photosynthesis, 100 kb away from the associated SNP because of the extensive LD in the associated genomic region. Based on our re-analysis of the data (see also below) and the examples from literature we are confident that *BSK3* is responsible for our observed marker-trait association.

i) Is the GWAS peak perhaps a ghost peak that is caused by multiple alleles? The Nordborg lab has recently published a nice case (Kerdaffrec et al 2016 eLife). Perhaps one could see if a multi-locus GWAS based on stepwise inclusion of markers could expose a similar phenomenon in this case (such as procedure is possible on the GWAPP portal). Also, a figure with the haplotypes in this region might be helpful to illuminate this issue.

Response: Thanks for this helpful suggestion. We carried out a new analysis using multi-locus mixed model (MLMM) and only identified the SNP previously detected in the original mixed linear model shown in Fig. 1b, suggesting that this unlikely refers to a ghost peak associated with multiple functional alleles. However, by carefully analyzing the linkage disequilibrium following the approach of Bac-Molenaar et al. (2015, Plant Cell), we found a long LD pattern in the associated region, which could account for the complex associations in this region. These new data are now shown in Supplementary Fig. 3 and described in the manuscript (lines 119-126): "To further resolve the multiple SNPs associated with this locus, we employed the multi-locus mixed model (MLMM) approach²⁸ which uses a stepwise model selection. The optimal model selected by this method identified the only SNP found in our initial GWAS on chromosome 4 (Supplementary Fig. 3). Estimates of the linkage disequilibrium (LD) between the top GWA SNP with surrounding markers revealed the presence of a long-range LD pattern which may account for the multiple significantly associated SNPs identified in this region."

ii) Perhaps the single trait GWAS was confounding? Would a multi-trait GWAS be more appropriate or a ratio of HN/LN traits for the analysis and expose the *BSK3* SNP?

Response: We agree that in some cases multi-trait GWAS might be superior to single trait GWAS, as exemplified by van Rooijen et al. (2017, Nat Commun) using principal components summarizing the variation of multiple traits or serial phenotypes across time as a phenotypic response. Although not hitting *BSK3*, using root length under LN/HN as trait is also interesting, but represents more the responsiveness of roots to N rather than their absolute extension under LN. Here, we targeted from the beginning primary root length under LN as major root trait for the GWAS approach.

iii) Might population structure correction interfere with the significance of the *BSK3* SNP? Perhaps the authors could consider running a simple linear model GWAS to check?

Response: As suggested, we performed an association scan with a linear model (Fig. 3 for reviewers) and found that our initial GWA SNP (in blue) was the most significantly associated SNP, while the SNP in the *BSK3* (in orange) showed moderately significant association. Hence, it is unlikely that overcorrection of population structure confounded the significance of the SNP in *BSK3*. However, as mentioned above, the causal SNP that results in *BSK3*'s amino acid substitution was not present in the 250 k SNP array. So far, we noted that the SNP detected from an association scan is not necessarily responsible for the functional polymorphism but rather indicates that the functional SNP in the causal gene is in strong linkage with the associated SNP. Here, we found that the causal SNP in *BSK3* is in strong LD ($r^2=0.7$) with the GWA-detected SNP, while it is absent from the 250 k SNP array.

iv) Can the authors really rule out that there is not a gene closer to the SNP and therefore more in line with the expected LD breakdown in Arabidopsis that is responsible for the GWAS peak? It could be a gene that is not contained in the RootMap (several thousand of genes are missing as it was ATH1 microarray based) or expressed below the detection level or expressed only transiently (diurnal or upon certain N conditions).

Response: To address the reviewer's concern, we now examined the primary root length of T-DNA knockout lines for all genes closer to the SNP (i.e., expected 20 kb LD window as usually taken in most GWA studies). The only exception was AT4G00891 for which no T-DNA line is available. We paid special attention to *ECA2*, in which the most significantly associated SNP is located, by examining two independent T-DNA insertion lines. The results showed that none of these insertion lines are defective in primary root growth irrespective of the N condition. The new results are now included in the revised Fig. 1c and Supplementary Fig. 4 and described in lines 126-139: "We assumed a confidence interval by computing the LD with surrounding markers ($r^2 > 0.7$) starting from 292979 to 398078, a region that includes 31 genes (Supplementary Table 3). For 16 of these genes, we could obtain T-DNA insertion lines that we then phenotyped in order to identify the causal gene underlying this locus. The top SNP was located in the gene *ER-TYPE Ca²⁺-ATPASE 2* (*ECA2*; AT4G00900). However, primary root length at both N conditions was similar to wild type in two independent *eca2* T-DNA insertion lines (Fig. 1c). An insertion mutant in the gene encoding for the *GSK3/Shaggy-like* kinase *ASKTHETA* (*ASKΘ*; AT4G00720) had shorter primary roots than its respective wild type, but this phenotype was independent of the N treatment (Fig. 1d). Interestingly, an insertion mutant of *BSK3* (AT4G00710) showed a conditional phenotype, with significantly shorter primary root than the wild type at LN, but not at HN (Fig. 1e). The primary root lengths of the remaining insertion lines were indistinguishable from wild type irrespective of the N condition (Supplementary Fig. 4)."

Since we could not rule out the possibility that we missed genes either not included in the ATH1 array or that were only transiently induced by low N, we decided to consider all genes contained in the computed confidence interval for which T-DNA lines were available. We amended the text to (lines 160-162): "Although we cannot fully rule out the possibility of the contribution of other genes at this locus, these independent approaches strongly indicated that *BSK3* is the causal gene for the observed marker-trait association found on chromosome 4 (Fig. 1b)."

v) In any case, the issues related to the large distance between top SNP and *BSK3*, as well as potential other explanations, should be discussed in the discussion and not be swept under the rug. I don't think discussing alternatives diminishes the story – it will rather increase its merit.

Response: Thanks for this recommendation. All issues raised in points i) to vi) are addressed as described above and now considered in the revised manuscript.

Figure 3 for reviewers. Regional Manhattan plot from an analysis using simple linear modeling. Red dots represent transformed $-\log_{10}$ p-values for the association between each marker and primary root length at low N. The SNP in *BSK3* (G1353A) is marked in orange, and the top SNP according to both, the mixed linear model and our new analysis using a multi-locus mixed model is marked in blue. The boxes on the horizontal axis indicate the position of the *AtBSK3* (in green) and *AtECA2* (in blue).

2. While being very interesting, the link to environmental parameters is not worked out well. First of all, it is not clear how the correlation analysis was conducted.

Response: We apologize for not having been sufficiently clear. The correlation analysis was carried out using Pearson correlation analysis as used previously by He et al. (2018, Nat Commun) for studying the relationship between NMR19-4 related phenotypes (i.e. DNA methylation, *PPH* expression and leaf senescence) and climatic variables. This has now been clarified in lines 579-580: "Pearson correlation was used to test associations between climate variables and primary root length". By also taking into account the comments of Reviewers 1 and 3, we decided to mention this aspect just briefly in the Discussion (lines 453-458), as this could represent an interesting starting point for future research.

In line 272 the authors state they relate haplotypes to the climate parameters. However, there is no hint how these haplotypes were computed, nor what they really are. Also, how does the GWAS top SNP and the *BSK3* SNP relate to this analysis.

Response: The term haplotype in line 154 of the revised manuscript refers to "protein coding haplotypes", as we follow the terminology from Chao et al. (2012, PLoS Genet) and Yang et al. (2018, Plant Cell). This has now been clarified in lines 516-522: "The *BSK3* protein haplogroups were analyzed using genomic data of 139 re-sequenced accessions that were phenotyped in the current experiment. Coding and predicted amino acid sequences were downloaded from *A. thaliana* 1001 Genomes Project (<http://signal.salk.edu/atg1001/3.0/gebrowser.php>). Sequences were aligned with ClustalW2.1 (<http://bar.utoronto.ca>). Only polymorphisms with minor allele frequency (MAF) >5% were considered."

Second, "This suggests that *BSK3* variants were potentially selected by precipitation." is worded very imprecisely. How can precipitation select something?

Response: Thanks for pointing this out. We now reformulated the sentence to (lines 464-466): "Future research will be necessary to test whether *A. thaliana* accessions may have adapted to topsoil drought by selection of the stronger *BSK3* allele allowing for longer primary roots."

Third, in the discussion the authors relate precipitation to drought, but precipitation can as well affect the leaching of nitrate by strong rainfalls or denitrification plus a huge array of other biotic and abiotic factors. This should be discussed.

Response: We agree with the reviewer that one could expect a positive association between precipitation and primary root length as plants with longer roots could potentially be better adapted to forage nitrate leaching into deeper soil layers as a consequence of more frequent and/or intense precipitation. However, we found that primary root length was negatively correlated with precipitation, pointing to topsoil drought rather than nitrate leaching as the relevant factor. In line with this scenario, nitrogen moves to the root surface mainly through mass flow for which water availability is of great importance. Therefore, less precipitation likely means that less nitrogen is

accessible for root uptake. Of course, these are rather preliminary speculations and future research will be required to more directly address this aspect.

Fourth, the method regarding the correlation is unclear. Please add detail on the linear model and what was done.

Response: The correlation analysis was carried out using Pearson correlation analysis as in He et al. (2018, Nat Commun). This has now been clarified in the text (lines 579-580).

Minor:

1. It would be nice to check the BR sensitivity of contrasting accessions for N-root responses and correlate this with the response to N; this could strengthen the case for causality of BSK3 further.

Response: We took up this suggestion and assessed the BR sensitivity of 56 natural accessions (29 accessions carrying the BSK3-P and 27 the BSK3-L variant). We found that accessions harboring the BSK3-L were more sensitive to BR as compared to those carrying the BSK3-P variant. We added these new results in Fig. 4d-e and Supplementary Table 5, and in the text (lines 225-228): "Further analysis of BR sensitivity in 56 natural accessions harboring either BSK3 allele revealed that accessions carrying the BSK3-L variant were more sensitive to the exogenous supply of BR than those with BSK3-P (Fig. 4d, e, Supplementary Table 5)." Moreover, we found that BR sensitivity in natural accessions was significantly correlated with primary root response to low N (new Fig. 5i). This result provided further support to our allelic complementation in which a significant correlation was observed between BR sensitivity and root responses to available N (Fig. 5h and Supplementary Fig. 12). We integrated this new evidence in the text (lines 271-275): "Furthermore, significant correlations were found between BR sensitivity, as determined by root length inhibition or hypocotyl elongation, and the responsiveness of primary root or average lateral root length to mild N deficiency (Fig. 5h, i; Supplementary Fig. 12), indicating a genetically determined coupling of BR response and root response to LN."

2. Line 309: The authors should outline the nature of the *bri1* mutants (is it loss of function?). Also, how can the authors reconcile the lack of phenotype in the *bri1* mutants with the function of BSK3 and BAK1 in the N-dependent response? That should be discussed.

Response: We now clarified this aspect in the text (lines 325-329): "Although exhibiting shorter primary and lateral roots than wild-type plants under both N conditions, two independent *bri1* loss-of-function mutants were still able to significantly increase primary root and lateral root length under LN (Supplementary Fig. 17b-f)."

Although BRI1 is not required for the N-dependent response, it might be possible that other BR receptors, such as BRL1 and BRL3, play a role in this process. Consistent with this notion, Fàbregas et al. (2013, Plant Cell) have shown that BRL3 and BRL1 use the same signaling components as BRI1 to modulate BR-dependent root growth. We now discuss this point in lines 425-429: "However, in our study, the root phenotypes of two *bri1* mutants indicate that a BRI1-independent mechanism modulates root foraging in response to N (Supplementary Fig. 17). This suggests that at LN other BR receptors, such as BRL1 and BRL3, which share conserved signaling with BRI1³⁴, could be activated to stimulate BR-dependent root growth".

3. Line 322 (and perhaps throughout the text): BSK3 is not causal for the observed variation. It is

most likely causal for part of the observed variation. Also, it would be nice if the authors could estimate somewhere how much of the phenotype the GWAS SNP or the BSK3 SNP explains.

Response: Thanks for this helpful suggestion. We now estimated the contribution of the GWAS SNP and found that it explains 11.7% of the observed variation for primary root length under LN. This result is not unusual for complex quantitative traits (e.g., Roojien et al., 2017, Nat Commun; Kalladan et al., 2017, PNAS; Bac-Molenaar et al., 2015, Plant Cell; Bac-Molenaar et al., 2016, Plant Cell Environ; Kooke et al., 2016, Plant Physiol). We integrated this information in the text, as follows (lines 116-119): “The locus on chromosome 4, in turn, contained 13 SNPs (FDR<0.1) and the most significantly associated SNP, which explained 11.7% of the observed phenotypic variation, was located at position 386,519 (Fig.1b, Supplementary Table 2).”

4. Line 332: change allelic variation to allelic variant. Also, are the authors sure there is only one variant?

Response: We agree and rephrased the sentence as follows (lines 348-352): “We nonetheless detected considerable natural variation in primary root length at LN among these accessions. Using GWAS, we could map two genes, *PHO1* and *BSK3*, which explained 6.3% and 11.7% of the observed variation, respectively (Fig. 1 and Supplementary Figs. 2 and 6).”

According to sequence data available in the 1001 genome database for 139 of all accessions used in this study, we do indeed find another non-synonymous SNP in *BSK3* leading to a P to T substitution at position 403. However, since only one out of 139 accessions has the *BSK3*-T variant, it is unlikely that this non-synonymous SNP is the causal one. In contrast, the C956T substitution leading to the L to P exchange at position 319 shows higher frequency (i.e. 105 L and 34 P as shown in Fig. 1h) and this particular amino acid substitution was strongly associated with the observed variation of primary root length under LN, as also reinforced by our allele swapping approach. In Fig. 1h (formerly Fig. 2d), we only indicate the most frequent SNPs. However, for the sake of completeness, we have now added a new Supplementary Table 4, describing all SNPs found in the *BSK3* gene in the 139 sequenced accessions.

Reviewer #3 (Remarks to the Author):

Major advances:

- GWA identifies *BSK3* and a non-synonymous SNP to be associated with variation in primary root length response in *Arabidopsis* accessions
- the first time that BR has been shown to be a component of regulation of root system architecture in response to differing levels of available N
- elegant analysis showing that this precise allelic variation contributes to variation in a variety of root system architecture traits, relative to Col-0

Response: Thanks for the positive comments about our work.

Major criticisms:

- the majority of data supporting this particular amino acid change in *BSK3* as being linked to variation in the LN, is via tests in Col-0. However, Col-0 is the accession that shows a mid-range trait, and the mutant phenotype of the *GSK3* is more similar to the accessions which do NOT show a LN response for primary root length (Petergof, Tsu-0 and Nd-1). Thus, I do not believe that the data provided

supports GSK3 as being broadly causal for the variation observed in the N-mediated changes in root system architecture.

Response: We agree with the reviewer. As mentioned in our response to Reviewer 2, we now estimated the contribution of the GWAS SNP and found it to explain 11.7% of the observed variation for primary root length under LN. We integrated this information in the text, as follows (lines 116-119): “The locus on chromosome 4, in turn, contained 13 SNPs (FDR<0.1) and the most significantly associated SNP, which explained 11.7% of the observed phenotypic variation, was located at position 386,519 (Fig.1b, Supplementary Table 2).”

Further, do Petergof, Tsu-0 and Nd-1 contain the P or the L haplotype in GSK3?

Response: According to the sequence available from the 1001 genome database, Petergof and Tsu-0 have a P-type BSK3 protein, while Nd-1 has the L-type. The reason why Nd-1 exhibits a shorter primary root despite carrying the L allele is likely due to the partial contribution of BSK3 to the natural variation of primary root length, as mentioned above. In fact, by following the suggestion of Reviewer 1, we found a second locus (on chromosome 3) with a false discovery rate close to 5% (q -value=0.06) explaining 6.3% of the phenotypic variation (Fig. 1b, Supplementary Table 2). The respective SNP is located in *PHO1*, a gene that has been previously shown to control natural variation of root allometry in response to phytohormones and nitrate (Rosas et al., 2013, PNAS). We then phenotyped the *pho1-2* mutant under our conditions and included the new data in Supplementary Fig. 2 and in the manuscript (lines 109-116): “At LN, we could map two major loci above a threshold of 10% false discovery rate (FDR) on chromosomes 3 and 4 (Fig. 1b, Supplementary Table 2). The associated SNP on chromosome 3, which accounted for 6.3% of the natural variation for primary root length at LN, was located in the *PHOSPHATE 1 (PHO1)*, a gene previously shown to modulate root allometry in response to auxin, ABA and nitrate²⁷. Compared to wild type, the *pho1-2* mutant failed to stimulate primary root elongation under LN (Supplementary Fig. 2), supporting the hypothesis that *PHO1* was the underlying gene for this locus.”

Why was the genomic complementation not done with the opposing allele in one of the most extreme accessions?

Response: We note that BSK3-P is not a loss-of-function allele as it can still complement the *bsk3,4,7,8* mutant (Figs. 3, 4 and 5, Supplementary Figs. 8 and 9). Since there is strong redundancy among BSK members, as also shown in our study (Fig. 2), it is expected to be difficult detecting the effect of an opposing allele even in extreme accessions. The *bsk3,4,7,8* mutant, on the other hand, offered an ideal background for transgenic complementation with either BSK3 protein variant.

-the sudden jump to exploring lateral root elongation and lateral root density is not supported by the original GWA screen. I suggest that the authors also refer to Gifford et al., 2013 in this respect.

Response: We investigated lateral roots besides primary root because we observed a significant decrease in average lateral root length (but not in lateral root density) in *bsk3* plants grown at LN as compared to wild-type plants (Supplementary Fig. 6). We believe that this phenotype is also relevant for the description of the overall effect of *BSK3* and BRs on N-dependent root architectural modifications. However, to better streamline the manuscript, we now moved the more detailed root architectural analysis to Supplementary Fig. 6 and amended the text as follows (lines 165-176): “A more detailed analysis of the root system architecture of *bsk3* plants demonstrated that the significantly reduced primary root length of these plants specifically when grown at LN was associated to their inability to stimulate root elongation in response to mild N deficiency

(Supplementary Fig. 6a, b). Furthermore, compared to wild-type plants *bsk3* plants had 13% shorter lateral roots but no differences in lateral root density at LN (Supplementary Fig. 6c, e). Nonetheless, as a consequence of the failed stimulation of primary root elongation and the partial loss of response of lateral roots to mild N deficiency, total root length of *bsk3* plants was only 78% of that of wild-type plants under LN (Supplementary Fig. 6d). “

Furthermore, with respect to the lateral root traits, differences in the magnitude of a response are being measured. Thus, pairwise statistical comparisons are not suitable and instead an ANOVA should be carried out looking at genotype x treatment interactions, followed by post-hoc Tukey tests. This should be done for all experiments involving lateral root traits.

Response: We agree with this comment and have carried out ANOVA followed by post-hoc Tukey tests to assess the difference among different genotypes in Figs. 2, 5, 7 and Supplementary Figs. 6, 7, 15, 16, 17. However, for the analysis of transgenic complementation assays, we need to apply a different statistical procedure, as now clarified in the section “Statistical analysis” of Methods (lines 584-590): “Root traits of different genotypes were compared by one-way ANOVA followed by post-hoc Tukey test at $P < 0.05$. However, for the analyses of transgenic complementation assays we conducted a Welch’s unequal variance t-test on ranked data. This was necessary because we had different numbers of individuals in each group and because basic assumptions for ANOVA and Tukey’s HSD test were violated due to Mendelian segregation within independent T2 lines and construct-dependent effects in different groups.”

-the fact that the P/L haplotypes regulate root length under all N treatments is also quite confusing. I appreciate that it does indicate that there is allelic variation, however, the link back to the original accessions that showed the most extreme responses is not made.

Response: We apologize for the confusion but note that the GWA association was found for absolute primary root length under low N and not for the response to low N (i.e., the ratio between LN and HN). However, during the course of this study, we also observed that *BSK3*-dependent BR sensitivity modulates not only the absolute length but also the root responses to low N. We have now made this point clearer in the manuscript by first presenting the GWA mapping and the initial mutant analysis in which primary root length at LN was assessed (Figure 1). We only introduce the root response to low N when describing the detailed analysis of *bsk3* plants (Figure 2), as it then became obvious that *BSK3* not only affects the absolute root length under low N but also the ability of plants to stimulate root elongation in response to low N. In addition, by assessing BR sensitivity in a subset of 56 natural accessions (29 with *BSK3*-P and 27 with *BSK3*-L, respectively), we found that those carrying the *BSK3*-L allele were more sensitive to this hormone and exhibited a significantly stronger primary root response to low N (Fig. 4d-e and Supplementary Table 5 in the revised manuscript). These new results provide additional support to our transgenic complementation approach by demonstrating that allelic variation in the *BSK3* coding sequence contributes to root responses to low N.

-the link with the precipitation is not strong and does not add to your story. There is no evidence in the literature for excessive wetness or flooding to be related to LN – this should either be tested or removed

Response: In principle, we agree with the reviewer. However, we think that the detected correlation represents an interesting starting point for future research. Considering further the comments of Reviewers 1 and 2, we now refer to these climate data briefly in the Discussion (lines 453-457):

“Consistent with this, we found that primary root length under low N significantly correlated with maximum precipitation in the wettest month of the natural habitats where the accession lines were sampled (Supplementary Fig. 18a). This raises the question as to whether the natural variation for this trait is, at least in part, related to water availability.”; and lines 464-466: “Future research will be necessary to test whether *A. thaliana* accessions may have adapted to topsoil drought by selection of the stronger BSK3 allele allowing for longer primary roots.”

Minor criticisms:

-a comparison of their associations relative to those in the Gifford et al. study would provide context for their identified SNPs and novelty

Response: As suggested, we compared our SNPs with those detected by Gifford et al. (2013, PLoS Genet) but found no overlap. This is not surprising because of differences in the N conditions used in both studies. Our mild N deficiency treatment stimulates root elongation, whereas the more severe N starvation used by Gifford et al. represses root growth.

-it should be stated in the main text what the N source is (NO₃ + NH₄)

Response: We now clarified this point in Results (lines 97-98): “After one week of pre-culture with sufficient N (11.4 mM total N supplied as 9.4 mM KNO₃ and 1 mM NH₄NO₃)” and in the Methods section (lines 478-481): “Seeds were surface sterilized in 70% (v/v) ethanol and 0.05% (v/v) Triton X-100 and sown on modified half-strength MS medium⁹ supplemented with 11.4 mM N (1 mM NH₄NO₃ + 9.4 mM KNO₃), 0.5% (w/v) sucrose, 1% (w/v) Difco agar (Becton Dickinson) and 2.5 mM MES (pH 5.6).”

- link between the accessions with no response, to the accessions with the greatest response, and the majority: in lines 111 and 112 – it should be clarified that this was only with a subset of the accessions and cannot necessarily be extrapolated to the whole population

Response: This question refers to the measurement of meristem size and cortical cell length in nine accessions shown in Fig. 1b-d of the original manuscript. We agree with the reviewer that the results obtained from this small subset cannot be extrapolated to the whole population. Considering that a detailed analysis of cellular traits in a much larger number of accessions is beyond the scope of this study and to address one point raised by Reviewer 4, we decided to remove these data from the revised manuscript. Instead, cortical cell length and meristem size measurements are now included for *bsk3* mutant lines in Fig. 2e,f, and Fig. 5e-g to define the cell biological process being affected by BSK3.

-is there any evidence for BR signaling as being regulated by N in the many transcriptional meta-analyses performed by G. Krouk and R. Gutierrez?

Response: Yes, we do find evidence for N-dependent regulation of some BR signaling genes in publicly available transcriptome datasets (see Table shown below). For example, BRI1 (AT4G39400) is among the 123 genes found to respond to systemic N signaling (Ruffel et al., 2011, PNAS); BKI1 (AT5G42750) has been shown to be responsive to nitrate in the study of Wang et al. (2007, Plant Physiol) and was identified in a gene cluster showing “N x accession” interaction by Gifford et al. (2013, PLoS Genet). In addition, BR biosynthesis genes like *CPD* and *DWF4* have also been shown to be responsive to N in transcriptome studies (Wang et al., 2003, 2004, Plant Physiol) and in the transcriptome meta-analysis carried out by Canales et al. (2014, Front Plant Sci). We integrated these evidences in the revised manuscript (lines 441-445): “Noteworthy, transcriptome studies have also

shown that genes involved with BR synthesis, such as *DWARF3/CONSTITUTIVE PHOTOMORPHOGENIC DWARF (DWF3/CPD)* and *DWARF4 (DWF4)*, respond to the plant's N status⁵³⁻⁵⁷. Thus, also BR biosynthesis may be targeted by N-dependent signaling.”

Gene	AGI	Reference	N-dependent response	
BRI1	AT4G39400	Ruffel et al., 2011	Regulated by systemic N signal	
BKI1	AT5G42750	Wang et al., 2007	Nitrate responsive	
		Gifford et al., 2013	Nitrogen responsive	
BZR1	AT1G75080	Wang et al., 2003	Upregulated by nitrate	
DWF4	AT3G50660	Canales et al., 2014	Responsive to N in meta-analysis of 27 N transcriptome datasets Ammonium and nitrate upregulated after 1.5h and 8 h	
		Patterson et al 2010		
		Widiez et al., 2011		Response to nitrate
		Wang et al., 2003		Upregulated by nitrate
		Wang et al., 2004		Upregulated by nitrate
DWF3	AT5G05690	Canales et al., 2014	Responsive to N in meta-analysis of 27 N transcriptome datasets	

-line 132 – this could be stated more clearly – that a gene needs to at least be expressed in one tissue in the root

Response: Following the suggestion of Reviewer 2, we have now assessed the phenotypes of available T-DNA insertion lines for the genes located within the estimated confidence interval.

-is there any GWA signal for the other BSK genes in your study or in the Gifford study?

Response: We carefully checked our data once more and could not find any other *BSK* gene close to the detected SNP or when extending the significance threshold to $-\log_{10}(P\text{-value}) = 4$. We also looked up the data from Gifford et al. (2013, PLoS Genet) and found no hints pointing to any of the *BSK* genes. As the outcome of GWAS is largely dependent on the allele frequency and linkage disequilibrium (LD), the non-detection of other *BSK* genes could be due to several reasons, such as a) insufficient allelic variation; or b) low LD of surrounding SNPs in the GWAS array with the functional SNP in BSKs, which could prevent the appearance of a significant GWA signal.

-lines 205 to 206 are not supported by the data

Response: We rephrased our statement more cautiously, as follows (lines 205-207): “Although our initial data indicated that an L to P substitution in the position 319 of the predicted kinase domain of BSK3 contributes to the natural variation of primary root length under LN (Fig. 1h, i), this evidence remained mainly based on correlations.”

Reviewer #4 (Remarks to the Author):

In this manuscript, Jia et al. showed that BSK3 regulates root foraging response under mild nitrogen deficiency condition. Authors analyzed primary root growth against nitrogen deficiency in 200 accessions of *Arabidopsis* and identified a genetic locus (BSK3) responsible for root foraging response from GWAS analysis. Authors insist that allelic variation (L319P) on BSK3 protein alters root elongation and BR sensitivity. Although authors tried to find genetic loci regulating root foraging

response with interesting screening approach, the current manuscript has serious problems and many issues to be further addressed.

<Manuscript organization and reliability>

1. Surprisingly, some images of the results are used in duplicate more than once. For example, phenotype images of *bsk3,4,7,8* in Figure 5a and Figure S4b are the same (but the scale is different). A more serious problem is that the authors used the same measurement in different experiments. Primary root length of *bsk3,4,7,8* (n=123) shown in Figure 5a are exactly the same as that of Figure S4c (*bsk3,4,7,8*, n=123) although they are independent experiments. In different experiments, authors seem to use the same control value. This is a serious problem which considered as a scientific misconduct.

Followings are other cases which I found in the manuscript.

- An image of #9 *proBSK3::BSK3-L/bsk3,4,7,8* line (Figure 5a) is found in Figure 6a.
- Two images (Col-0 and *bsk3,4,7,8* treated by BL) shown in Figure 6a is duplicated in Figure S5a. In addition, the same values of primary root length were used in both Figure 5b and Figure S5b.
- #6 (*proBSK3::BSK3-L/bsk3,4,7,8*) line of Figure S4b is duplicated in Figure 6a.

Response: We apologize for not having indicated clearly some of the experimental details in the legends of the original manuscript. This could have avoided confusion. In order to compare the two promoter variants and the two coding variants, we conducted an extremely large experiment in which we grew wild type (Col-0), *bsk3,4,7,8* mutant as well as individual allelic complementation lines carrying either *proBSK3_{Col-0}::BSK3-P*, *proBSK3_{Col-0}::BSK3-L* and *proBSK3_{Cvi-0}::BSK3-L* at both mock (-BL) and BR supplied (+BL) agar. In this experiment, we addressed two major questions:

Q1) Is root growth associated with *BSK3* coding or promoter variants?

Q2) Is BR sensitivity affected by *BSK3* promoter variants or by *BSK3* coding variants?

According to our results, variation in the coding sequence of *BSK3* is relevant for both, root growth and BR sensitivity (data shown in original Fig. 5 and 6, Supplementary Figs. 4 and 5). However, the original figure containing all genotypes and the two treatments became very big. Therefore, we decided to split the results into Fig. 5 and Supplementary Fig. 4 to address Q1, and Fig. 6 and Supplementary Fig. 5 to address Q2. In favor of using the absolute values instead of normalized data for the BR sensitivity assay, the previous version showed the same values for mock (-BL) treatment in Fig. 6 and Supplementary Fig. 5.

After the reviewer's comments, we realized that this procedure can cause misunderstanding.

Therefore, in the revised version we present the results of a completely new experiment carried out with wild type (Col-0), *bsk3,4,7,8* and allelic complementation lines expressing *proBSK3_{Col-0}::BSK3-P*, *proBSK3_{Col-0}::BSK3-L* or *proBSK3_{Cvi-0}::BSK3-L* to specifically address Q1 (new results shown in Fig. 3 and in Supplementary Fig. 10), while the original dataset is now only used to address Q2 (new Fig. 4a-c and Supplementary Fig. 11). Thereby we do not present the same controls in different figures.

However, to highlight the relevance of coding variants for natural variation of root growth, we have to show the data for coding variants and promoter variants separately. We now clarify this point in the legend of Supplementary Fig. 10 and Supplementary Fig. 11. In addition, as suggested by the reviewer, we now show phenotypic values for individual transgenic lines along with the averaged values of all lines for each construct (see more below).

2. Although authors used multiple transgenic lines (usually three independent lines) and showed their phenotypes, in measurement, authors seem to combine the values all of the lines and present

their mean as a single transgenic line. Unlike controls, error bar is too big although n value is very high. It is doubtful that three independent lines showed a different pattern for root growth.

Response: We apologize for not clearly stating this information in the original version. In Fig. 5b-c and Fig. 6b-c, we used 14 independent lines and in Fig. 6e-f 17 independent lines. This information is now clearly indicated in the legends of these figures (Figs. 3 and 4 in the revised version). We used a large number of independent lines for each transgene in order to best avoid the random effect of differential expression caused by the different T-DNA integration sites in each independent line. A similar approach has been used in many previous studies (Satbhai et al., 2017, Nat Commun; Meijón et al., 2014, Nat Genet; Anwer et al., 2014, eLife; Jiménez-Gómez et al., 2010, PLoS Genet; Beuchat et al., 2010, PNAS). In the revised manuscript, we show the average values for each individual line and also the average value of all independent lines assessed for each construct (new Fig. 3 and Supplementary Fig. 10).

3. In all experiments using transgenic plants expressing BSK3, authors should compare BSK3 protein level. Regardless of the native promoter or 35S-promoter, gene expression is affected by the insertion position and copy number of T-DNA on the chromosome. Authors can compare BSK3-L and BSK3-P activity based on the similar protein level.

Response: We agree that in some cases assessing BSK3 protein levels is more reliable than gene expression. This is also why we used a very large number of independent transgenic lines for each construct (14 for the native promoter and 17 for the 35S promoter). Nevertheless, to further address the reviewer's concern, we generated new transgenic lines expressing *proBSK3_{Col-0}::BSK3-P-GFP* or *proBSK3_{Col-0}::BSK3-L-GFP* in the *bsk3,4,7,8* mutant background. We then took GFP fluorescence intensity as a proxy for BSK3 protein abundance to select three independent lines for each BSK3 variant with comparable variation in protein abundance. Analysis of these lines on -BL and +BL confirmed that the BSK3-L variant restores BR sensitivity in the *bsk3,4,7,8* more efficiently than the BSK3-P variant. These new results are now shown in Supplementary Fig. 8.

Furthermore, to assess BSK3-dependent BR sensitivity independently of a transgenic approach, we grew 56 natural accessions (27 accessions carrying BSK3-L and 29 lines carrying BSK3-P variant) on -BL and +BL. Also in this experiment we found evidence that plants are more sensitive to BR when carrying the BSK3-L variant (Fig. 4d-e, Supplementary Table 5 in the revised manuscript). These new datasets are also described in the text (lines 221-230): "To control for variation at the protein level, we introduced translational fusions between the two BSK3 protein variants and GFP into *bsk3,4,7,8* plants. Both variants recovered BR sensitivity of the quadruple mutant, but complementation was more efficient with BSK3-L-GFP irrespective of variations in GFP fluorescence intensity (Supplementary Fig. 8). Further analysis of BR sensitivity in 56 natural accessions harboring either BSK3 allele revealed that accessions carrying the BSK3-L variant were more sensitive to the exogenous supply of BR than those with BSK3-P (Fig. 4d, e, Supplementary Table 5). Taken together, these results showed that BSK3-L is more efficient than BSK3-P in mediating BR signaling."

4. Authors investigated root foraging responses of 200 accession of Arabidopsis under low N condition (Supplemental Table 1), selected 9 ecotypes, and measured meristem size, cortical cell length. Why did the authors show only 3 ecotypes' meristem images although authors showed all 9 ecotypes' cortical cell length? In addition, authors need to show LN response of 9 ecotypes in main figure (prior to Figure 1b).

Response: Since our quantitative analysis indicated that low N can differentially target meristem size (Supplementary Fig. 2a of original manuscript) and mature cell length (Fig. 1c of original version)

depending on the accession being investigated, we showed in the original manuscript images of meristem size and mature cortical cell length for only three of the original nine accessions, as they represented the effect of N on meristem size, cortical cell length or both (Fig. 1b in the original version). For the revised version, we performed the analysis of cellular traits when comparing Col-0, *bsk3* and *bsk3,4,7,8* (Fig. 2d-f), when assessing the effect of the complementation of *bsk3,4,7,8* with the two BSK3 protein variants (new Fig. 5e-g), and when comparing *Ws-2* and *bak1-1* (new Supplementary Fig. 15). This more straightforward approach allowed us removing the data for nine natural accessions in the original manuscript (formerly Fig. 1b-d).

5. From Figure 1, authors divided 9 accessions into three categories based on their root foraging response to low N condition. However, their responses are not tightly correlated with a.a. 319 variation of BSK3. Weak/no: Petergof (P), Tsu-0 (P), Nd-1 (L); intermediate: Co(P) and Col-0 (L); strong: LDV-14 (NA), LDV-58(NA), Do-0(P) and Cvi-0(P). How can authors explain this contradiction? Response: We agree with the reviewer that in this set of accessions the primary root response does not correlate well with BSK3's amino acid variation at position 319. In the original manuscript, we intended to address which cellular trait (i.e., meristem size and/or cell elongation) was altered by low N to cause the differential primary root length at low N that we observed in the 200 accessions. Therefore, accessions with contrasting primary root responses were selected for this analysis even before we had mapped *BSK3* for primary root length under low N. We have now realized that this is confusing, because it raises the expectation that the primary root response correlates with the BSK3 variants. However, the GWA association was found for absolute primary root length under low N and not for the response to low N (i.e., the ratio between LN and HN). In addition, as mentioned in our responses to Reviewers 2 and 3, the GWA peak linked to *BSK3* explains 11.7% of the primary root length variation at low N, indicating that other genes contribute to the variation in this trait. This is expected for a complex trait. As shown in the Manhattan plot (Fig. 1b), a second GWA peak also pops out and contributes to a part of this variation (6.3%). Therefore, it is conceivable that in the whole population some accessions harboring the BSK3-P allele may exhibit a strong primary root response to low N due to the effect of other relevant alleles. We note that in terms of absolute primary root length, accessions carrying the BSK3-L allele had significantly longer primary roots under low N (not response to low N) than those carrying the BSK3-P allele (Fig. 1i).

In order to accommodate the new datasets and to streamline the manuscript, the revised version presents the analysis of cellular traits when comparing Col-0, *bsk3* and *bsk3,4,7,8* (Fig. 2d-f), when assessing the effect of the complementation of *bsk3,4,7,8* with the two BSK3 protein variants (new Fig. 5e-g), and when comparing *Ws-2* and *bak1-1* (new Supplementary Fig. 15). Moreover, as a detailed analysis of cellular traits in a much larger number of accessions is beyond the scope of this study, we removed the data of nine accessions shown in the original Fig. 1b-d.

Later, authors' experiments were just focused on Col-0 carrying BSK3-L but mild response to low N. Authors compared promoter activity between Col-0 and Cvi-0. However, Cvi-0 showed strong LN-induced root elongation although it contains BSK3-P. It is difficult to understand the logic to design the overall experiment.

Response: Please note that *BSK3* expression in natural accessions did not correlate significantly with primary root length (Fig. 2c in original version, Fig. 1g in the revised version), suggesting that expression variation caused by polymorphism in the promoter is not associated with primary root length variation. To further verify this conclusion, we selected the promoter of Cvi-0 and Col-0, two accessions differ significantly in primary root length at low N. We hypothesized if this phenotypic

difference is associated with promoter variation, then expressing the same BSK3 variant (here we selected BSK3-L) under the control of *BSK3* promoters from either Col-0 or Cvi-0 will lead to differential complementation of root growth and BR sensitivity. However, the results shown in Supplementary Figs. 4 and 5 in the previous version (Supplementary Figs. 10 and 11 in the revised manuscript) indicate that promoter variants do not significantly affect the complementation efficiency of *bsk3,4,7,8* in terms of root growth and BR sensitivity. These results further strengthen our conclusion that the variation in coding sequence of BSK3 is relevant for the primary root length variation at low N.

<This reviewer could not find any concrete evidence that BR regulates LN-induced root growth>

1. In Figure 2e, although primary root length of L type BSK3 was slightly longer than P type BSK3 in HN as well as LN condition, the ratio of HN/LN between L type and P type look almost similar (regardless of the amino acid type, root elongation ratio in response to low N is similar). This means that variation of BSK3 sequence is not major reason for different foraging response.

Response: As stated above, please note that the GWA association was found for primary root length under low N rather than the response to LN (i.e., the ratio of LN/HN). Therefore, it is not surprising that the ratio (response) does not significantly differ between two protein haplotypes. By contrast, the significant difference of primary root length between BSK3 protein haplotypes under low N provides strong evidence that BSK3 is causal for the identified GWA peak. Taking into account the complex nature of the trait and the existence of additional genetic contributors as detailed in one of our previous answers, it is conceivable that in the whole population some accessions with the BSK3-P allele may also show a strong response due to compensation effects by other alleles.

2. Authors described that cortical cell size correlates with strong response for LN condition. It is known that BR promotes root elongation in epidermal layer. Is there any specific function of BR in cortical cells of root? Authors need to explain the BR action in cortical cells.

Response: We agree with the reviewer that BR perception in epidermal cells has been shown to regulate root growth (Hacham et al., 2011, Development). In our study, we measured the length of mature cortical cells as an indicator for cell length; this approach has been practiced also in other studies (Hacham et al., 2011, Development; Meijón et al., 2014, Nat Genet; Fridman et al., 2014, Genes & Dev). Assessing the cell-type specificity of BR responses to mild N deficiency is beyond the scope of our study.

3. Figure 6d and e, authors need to test BR sensitivity for 35S-BSK3-P or 35S-BSK3-L.

Response: As genetically enhanced BR signaling inhibits root elongation (Gonzalez-Garcia et al., 2011, Development), also our transgenic lines overexpressing BSK3-L exhibited significantly shorter primary root and slower growth rate compared to those overexpressing BSK3-P even in the absence of exogenously supply BL (Fig. 6d-f in previous version, Fig. 4f-h in the revised manuscript). This result already suggested that BSK3-L is more efficient than BSK3-P in relaying the BR signal. However, we took up the reviewer's suggestion and performed a new experiment to test the BR sensitivity of our lines. We observed that overexpression of BSK3-L induced a stronger BR response, as reflected by significantly shorter primary root length and increased hypocotyl elongation when supplied with BL. These results are included in the new Supplementary Fig. 9 and described in the text (lines 237-240): "In addition, transgenic lines overexpressing the BSK3-L variant exhibited significantly stronger BR sensitivity compared to those overexpressing BSK3-P (Supplementary Fig. 9)."

4. Based on the results of complementation experiments, authors argued that BSK3-L is more functional in BR signaling and root foraging response than BSK3-P. What is the mechanism underlying different response? This variation affects on interaction of BSK3 with known BR signaling component or kinase activity?

Response: Even though BSK3 was first described more than 10 years ago (Tang et al., 2008, Science), its biochemical activity still remains elusive. Whereas the involvement of BSKs in BR signaling as activators of BSU1 has been clearly demonstrated (Kim et al., 2009, Nat Cell Biol; Zhang et al., 2016, Plant Physiol; Ren et al., 2019, PLoS Genet), as discussed in the manuscript (lines 376-388), no kinase activity could be consistently demonstrated so far (Kim et al., 2011, Mol Cell; Grutter et al., 2013, J Mol Biol). A study published during the revision of this manuscript (Ren et al., 2019, PLoS Genet, in press, <https://doi.org/10.1371/journal.pgen.1007904>) further reinforced the importance of BSK3 in BR signaling. Interestingly, the new study also found no evidence for kinase activity although demonstrating that the predicted kinase domain is necessary for BSK3 function.

Therefore, we don't know how we should compare the two BSK3 variants, as a meaningful protein activity assay is not yet feasible. As replied to Reviewer 1, attempts to gain insights by modeling the two BSK3 proteins with the partial structure of BSK8 produced inconclusive results, as the BSK8 structure ends just close to our relevant amino acid residue. Thus, to directly assess the effect of the L to P substitution at position 319, it will be necessary to a) obtain a complete crystal structure of BSK3; b) extend the search for BSK3 interaction partners beyond the known BR signaling regulators; and c) to design a method to allow comparing (in quantitative terms) the proposed scaffold function of BSK3 (Ren et al., 2019) for the two BSK3 protein variants found in our study. As these presumptions are currently not given, we are unable to describe the mechanistic action of the P/L substitution in BSK3.

5. In Figure 5 and 6, current results just partly support that BSK3-L form is more functional in BR signaling. Have authors tested BL or Brassinazole effect on LN-induced root elongation?

Response: As suggested by the reviewer, we tested the effect of BL and Brassinazole (BRZ) on the root growth of Col-0 plants under HN and LN (Fig. 4 for reviewers). We observed that a low concentration of BL (0.25 nM) in the form of 24-epibrassinolide significantly stimulated primary root and lateral root growth at HN as compared to -BL (Fig. 4a-c for reviewers). At LN, this same concentration had either no effect or was even inhibitory. The supply of BRZ, which blocks the activity of the enzyme DWF4 catalyzing the rate-limiting step of BR biosynthesis, strongly repressed the response (i.e. LN/HN) of primary root and lateral roots to N (Fig. 4d-f for reviewers). These results suggest that LN not only enhances BR sensitivity, but may also increase BR biosynthesis.

As replied in more detail to Reviewer 3, we also found evidence in published transcriptome data that N deficiency affects the expression of many genes involved with BR synthesis and signaling. We integrated this piece of evidence in the revised manuscript (lines 441-445): “. Noteworthy, transcriptome studies have also shown that genes involved with BR synthesis, such as *DWARF3/CONSTITUTIVE PHOTOMORPHOGENIC DWARF (DWF3/CPD)* and *DWARF4 (DWF4)*, respond to the plant's N status⁵³⁻⁵⁷. Thus, also BR biosynthesis may be targeted by N-dependent signaling.”

Figure 4 for reviewers. Root growth of Col-0 in response to BL and BRZ under different N availabilities. Seven-day-old wild type (Col-0) seedlings were pre-cultured on 11.4 mM N and then transferred to solid agar media containing either high N (HN, 11.4 mM N) or low N (LN, 0.55 mM N) in the presence or absence of indicated concentrations of BL (24-epibrassinolide) or BRZ (Brassinazole). Root system architecture was assessed after 9 days. **(a-c)** Appearance of plants **(a)**, primary root length **(b)** and average lateral root length **(c)** of Col-0 plants grown under BL supply. **(d-f)** Appearance of plants **(d)**, primary root length **(e)** and average lateral root length **(f)** of Col-0 plants grown in absence or presence of BRZ. Bars represent means \pm s.e. ($n = 13-15$ plants). Different letters indicate significant differences at $P < 0.05$ according to one-way ANOVA and post-hoc Tukey test within respective N treatment (in **b** and **c**) or across N and BZR treatments. Asterisks indicate statistically significant differences between two N treatments at individual BR concentration (in **b** and **c**) according to Welch's t test ($*P < 0.05$, $**P < 0.01$, $***P < 0.001$, ns, not significant). Scale bars, 1 cm.

6. Authors suggested that low N activates BR signaling through elevation of BAK1 mRNA level. BAK1 is multifunctional co-receptor which interacts with several RLKs. the phenotype of *bak1* mutant which shows a subtle difference (Figure 9) can't support that BR regulates N deficiency response of roots. In addition, BSK members also are regulated by another RLKs such as FLS2.

Response: We agree with the reviewer that *bak1-1* plants show a subtle primary root response (Fig. 9 in previous version, Fig. 7 in the revised version). However, the effect was statistically significant (LN/HN for primary root length = 1.14 ± 0.01 for WT vs 1.06 ± 0.02 for *bak1-1*, $p=0.007$, Welch's t test). The effect of *BAK1* mutation was more obvious for the response of lateral roots (LN/HN for average LR length = 2.22 ± 0.06 for WT vs 1.55 ± 0.06 for *bak1-1*, $p=3.8e-9$, Welch's t test). Together, these differences suggest that *BAK1* is indeed involved with the regulation of root responses to low N. The rather small significant difference for the primary root could be due to redundancy as it has been shown that other *BAK1* homologs can function in BR signaling (Jeong et al., 2010, Mol Cell; Karlova et al., 2006, Plant Cell; He et al., 2007, Curr Biol; Gou et al., 2012, PLoS Genet).

Regarding the role of BAK1 as a multifunctional co-receptor in different physiological and development process, it is unlikely that the observed phenotypes were due to BAK1's role in plant innate immunity because we grew our plants in sterile conditions. We did also not observe severe growth retardation or symptoms of spontaneous cell death in *bak1-1* as those reported for the *bak1 bkk1* double mutant (He et al., 2007, Curr Biol). So far only BSK1 has been shown to form a complex with FLS2 and to function in the flg22-elicited plant innate immunity response, as knock-out lines for other BSKs, including *BSK3*, showed no immunity-related phenotype (Shi et al., 2013 Plant Cell). To further test our conclusions and to address the reviewer's concerns, we overexpressed *BSK3* in the *bak1-1* mutant. Interestingly, our new results show that ectopic expression of *BSK3*, which can enhance BR signaling in *bsk3,4,7,8*, can also largely restore the BR sensitivity and the low N-dependent root response of *bak1-1* plants. These data were included in Fig. 7e-g and Supplementary

Fig. 16 and described in the text (lines 317-324): “BAK1 is involved in multiple signaling pathways³⁴. To test whether impaired N deficiency-induced root growth of *bak1-1* plants was indeed due to perturbed BR signaling, we overexpressed *BSK3* in the *bak1-1* mutant. Ectopic expression of *BSK3* was able to largely restore the sensitivity of *bak1-1* plants to exogenous BR (Supplementary Fig. 16). Most importantly, overexpression of *BSK3* restored significantly both the primary and lateral root response of *bak1-1* to LN (Fig. 7e-g), corroborating that the reduced root response of *bak1-1* plants to LN is due to impaired BR signaling.”

Most of all, Figure S8 indicates that *bri1* does not show any difference in LN-induced root growth. BSKs are activated by BRI1 but not by BAK1. If the phenotype of the *bsk3* mutant was due to the defect of BR signaling, the *bri1* mutant must show the same phenotype.

Response: We disagree with the reviewer’s assumption that the *bri1* mutant must show a low N-dependent phenotype. Although BSKs are activated by BRI1, it has been shown that ectopic expression of *BSK3* in null allele *bri1-116* could suppress its dwarf phenotype (Tang et al., 2008, Science). Moreover, during preparation of the revised manuscript, a new study also showed that enhanced expression of *BSK3* in the null mutant *bri1-801* could recover its dwarfism, root growth, shoot growth as well as male fertility (Ren et al., 2019, PLoS Genet). These experiments clearly indicate that *BSK3* can activate BR signaling also in absence of a functional BRI1 receptor. In Arabidopsis, also BRL1 and BRL3 can interact with BAK1 *in vivo* and mediate BR-dependent root growth (Fàbregas et al., 2013, Plant Cell). Even more importantly, in the study of Fàbregas et al. (2013, Plant Cell), of all BSKs, only BSK1 and BSK3 were shown to co-immunoprecipitate with BRL3. Thus, it might not be surprising if BRL3 (or even BRL1) functions in a BR-mediated pathway that integrates environmental cues with general growth processes. However, we think that this aspect must be addressed in a separate study that focusses on combinatorial aspects among BRI/BRL, BAK and BSK proteins and their implication in root growth. Nonetheless, we discussed this point in our manuscript (lines 425-429): “However, in our study, the root phenotypes of two *bri1* mutants indicate that a BRI1-independent mechanism modulates root foraging in response to N (Supplementary Fig. 17). This suggests that at LN other BR receptors, such as BRL1 and BRL3, which share conserved signaling components with BRI1³⁴, could be activated to stimulate BR-dependent root growth.”

Reviewers' comments:

Reviewer #1 (Remarks to the Author):

I enjoyed reading the revised manuscript and find that that the authors have done a good job of responding to all comments, including those for discussion rather than action. Inclusion of more careful explanation, and of detail to develop points has been well done. I did have a couple of comments about the new data provided to reviewers though.

(1) Shoot N concentration measurements (Figure 2 for reviewers) – the quadruple mutant shows reduced N in the shoot and this is not rescued in the P or L complemented lines (except in line P-13) which seems rather bizarre. The explanation given for this does not make sense since it talks about the wider implication of altered N uptake/storage, not the result shown in the graph. Would the authors not expect to see complementation of this trait?

(2) I would include 'Figure 4 for reviewers' within the manuscript since it supports the BL effect at LN.

Acknowledgement for the homology modelling is added which is nice, but seeing as there was no clear conclusion from this, and it is not included, this could be confusing for readers.

Reviewer #2 (Remarks to the Author):

In the manuscript "Natural variation of BSK3 tunes brassinosteroid signaling to regulate root foraging responses to low nitrogen" Zhongtao Jia and colleagues uncover the role of Brassinosteroid signaling and in particular the gene BSK3 for low-Nitrate induced root growth foraging patterns. They further show that natural variants of BSK3 are involved in shaping observable natural variation of root growth responses to low N. Interestingly, they also expose a correlation of BSK3 allele distribution and precipitation parameters, hinting towards a potential link of water availability and the impact of BSK3 on N related root growth responses. This is an impressive and important work that uncovers new and exciting biology, as well as provides potential for meaningful agricultural application as improving nitrate related root foraging traits might improve crop yields and reduce inputs and environmental damage.

The manuscript is written very well and in a concise manner. The research is of high quality and the scope of the work is impressive. The conclusions are supported by the data. I truly enjoyed reviewing this manuscript.

While I had voiced multiple concerns upon reviewing the last version of the manuscript, the authors have done an excellent job to address all my concerns. Their revisions have enhanced an already impressive manuscript substantially. I commend the authors to this work.

Reviewer #5 (Remarks to the Author):

The authors have addressed many of the previous reviewers concerns.

In general root foraging is about the total space that the root system occupies - why is total root system size not analyzed for all genotypes? The title and general approach to the discussion of BSK3 should be more focused on the primary root growth under low N conditions.

Major Concerns:

Line 187 - This statement needs to be clarified to say that the effects observed are no different

than bsk3 single mutant. The way it reads now is that these genotypes are not different from Col-0 control.

Line 219 - Are these plants restored to Col-0 levels? Why is there is no comparison back to the WT control for the mutants?

Line 313 - What is the hypothesis behind BAK1 taking 6 days to be transcriptionally responsive to N conditions? This is an incredibly delayed response.

Line 484 - This selection of similar sized seedlings is biasing the sampling - if there is variation in the growth at early stages due to the mutants, this is nullifying any of the impact of those mutations. Please justify why this approach was taken.

Figure 3b - Why are these plants are at a different age than the previous data? The authors' response to reviewers is justified but this needs to be included in the manuscript. I would like to see which of the individual lines are statistically different from the quadruple mutant. Also, why are these lines not compared to Col-0? A better presentation of the data would sort the lines by root length measurement to see what the distribution of the lines actually is. This would increase the argument that the P allele lines comprise more of the shorter primary roots measured. The current organization is difficult to interpret.

Figure 4 - B and C - it is unclear which letters are associated with which groups. Why are all categories not included in the ANOVA and posthoc Tukey testing?

Figure 4 - G and H need to be analyzed using an ANOVA and post-hoc Tukey test to determine statistical significance between all the genotypes.

The shoot N concentration needs to be added as a supplemental figure to the paper. This data is important as the authors claim that increasing root size will have greater yield, and this suggests that that assumption is not always the case. Not including this data is misleading.

Minor Comments:

Line 48: In the split root assays there are no primary roots - all the roots studied are lateral roots - secondary or tertiary

Line 173: This statement should be altered to state that the transgenic rescue is not statistically significantly similar to WT.

Line 185: For Figure 2 make a plot with total root length - this is addressed in the discussion of this figure but no combined data is provided.

Line 278 - Specify that the "root foraging" response is for primary roots only.

Reviewers' comments:

Reviewer #1 (Remarks to the Author):

I enjoyed reading the revised manuscript and find that that the authors have done a good job of responding to all comments, including those for discussion rather than action. Inclusion of more careful explanation, and of detail to develop points has been well done. I did have a couple of comments about the new data provided to reviewers though.

Response: We thank the Reviewer for the encouraging comments.

(1) Shoot N concentration measurements (Figure 2 for reviewers) – the quadruple mutant shows reduced N in the shoot and this is not rescued in the P or L complemented lines (except in line P-13) which seems rather bizarre. The explanation given for this does not make sense since it talks about the wider implication of altered N uptake/storage, not the result shown in the graph. Would the authors not expect to see complementation of this trait?

Response: Root architecture is only one factor that can influence N capture and shoot N accumulation. The reason why shoot N levels of the *bsk3,4,7,8* mutant could not be complemented by any of the BSK3 variants may be manifold, but we suspect a major impact by non-redundant roles of the BSK homologs. It has been shown that the ammonium and nitrate transporters *AMT1;1* and *NRT1;1*, respectively, are regulated by BR-responsive transcription factors (Sun et al., 2016, Dev Cell; Xuan et al., 2017, J Exp Bot), indicating that BR signaling is also involved in the physiological regulation of N uptake in roots. Therefore, BSK4, BSK7 and/or BSK8 might be involved in BR-dependent regulation of N transporters in roots.

Since Reviewer #5 asked us to include these data in the manuscript, these results are now presented in Supplementary Fig. 13 and described in lines 279-284: “We then tested whether the superior root length brought about by the L-allele also led to elevated N uptake. While we found that under LN supply *bsk3,4,7,8* accumulated significantly less N in the shoot, complementation with either BSK3 variant did not restore shoot N accumulation (Supplementary Fig. 13). This failure may be due to additional, non-redundant functions of BSK proteins and of BRs in the transcriptional regulation of N transporters^{45,46}.”

(2) I would include ‘Figure 4 for reviewers’ within the manuscript since it supports the BL effect at LN.

Response: Thanks for this suggestion. These results are now shown in Supplementary Fig. 7 and described in lines 179-184: “We found that both primary and lateral roots of N-deficient plants were hypersensitive to exogenous supply of the bioactive BR 24-epibrassinolide (BL) (Supplementary Fig. 7a-c). Furthermore, the supply of the BR biosynthesis inhibitor brassinazole largely prevented the low N-induced elongation of primary and lateral roots (Supplementary Fig. 7d-e), suggesting that BRs play a role in root system architectural changes under LN.”

Acknowledgement for the homology modelling is added which is nice, but seeing as there was no clear conclusion from this, and it is not included, this could be confusing for readers.

We agree with the reviewer and now only acknowledge Jochen Balbach for valuable discussions.

Reviewer #2 (Remarks to the Author):

In the manuscript “Natural variation of BSK3 tunes brassinosteroid signaling to regulate root foraging responses to low nitrogen” Zhongtao Jia and colleagues uncover the role of Brassinosteroid signaling and in particular the gene BSK3 for low-Nitrate induced root growth foraging patterns. They further show that natural variants of BSK3 are involved in shaping observable natural variation of root growth responses to low N. Interestingly, they also expose a correlation of BSK3 allele distribution and precipitation parameters, hinting towards a potential link of water availability and the impact of BSK3 on N related root growth responses. This is an impressive and important work that uncovers new and exciting biology, as well as provides potential for meaningful agricultural application as improving nitrate related root foraging traits might improve crop yields and reduce inputs and environmental damage.

The manuscript is written very well and in a concise manner. The research is of high quality and the scope of the work is impressive. The conclusions are supported by the data. I truly enjoyed reviewing this manuscript.

While I had voiced multiple concerns upon reviewing the last version of the manuscript, the authors have done an excellent job to address all my concerns. Their revisions have enhanced an already impressive manuscript substantially. I commend the authors to this work.

Response: We thank the Reviewer for the positive comments.

Reviewer #5 (Remarks to the Author):

The authors have addressed many of the previous reviewers concerns.

In general root foraging is about the total space that the root system occupies - why is total root system size not analyzed for all genotypes? The title and general approach to the discussion of BSK3 should be more focused on the primary root growth under low N conditions.

Response: We agree with the Reviewer that root foraging is related to the total soil volume explored by the roots and thus best reflected by total root length in our study. Since total root length is determined by the length of the primary root and all lateral roots as well as the number of lateral roots, the genetic architecture underlying this trait is even more complex than the individual traits contributing to total root length. Even though we mapped *BSK3* by GWAS using primary root length only, our subsequent phenotypic characterization of the *bsk3* mutant as well as our transgenic complementation approach demonstrated that BSK3-mediated BR signaling and allelic variants of BSK3 do not only modulate the N-deficiency response of the primary root but also that of lateral roots and total root length (Figs. 2 and 5, Supplementary Figs. 6 and 8). Based on these observations we consider it justified to relate the natural variation in BSK3 to root foraging, as we did in the title. Nevertheless, whenever formulating in more detail, we referred to the function of BSK3 in primary root elongation.

Major Concerns:

Line 187 - This statement needs to be clarified to say that the effects observed are no different than *bsk3* single mutant. The way it reads now is that these genotypes are not different from Col-0 control.
Response: Thanks. To make it clearer, we now amended the text to (lines 192-194): “Relative to *bsk3*, only minor effects were observed for *bsk3,4* double and *bsk3,4,7* triple mutants at HN condition, while no effect was observed for *bsk3,4* double, and *bsk3,4,7* or *bsk3,4,8* triple mutants at LN.”

Line 219 - Are these plants restored to Col-0 levels? Why is there is no comparison back to the WT control for the mutants?

Response: In the experiment shown in Fig. 3, our main goal was to assess whether the two BSK3 variants could differentially stimulate primary root elongation taking advantage of the short root phenotype of the *bsk3,4,7,8* quadruple mutant. Since we only reintroduced one of the four BSKs knocked out in this quadruple mutant, we did not expect to fully restore primary root length back to wild type. Therefore, in this experiment, we did not make direct comparisons to Col-0 but only compared the BSK3-L/P-complementing lines to *bsk3,4,7,8* and the two BSK3 protein haplotypes with each other.

Line 313 - What is the hypothesis behind BAK1 taking 6 days to be transcriptionally responsive to N conditions? This is an incredibly delayed response.

Response: We attribute this apparent delay to our experimental setup. Please note that in our growth system, plants were always pre-cultured under sufficient N for seven days before being transferred to HN or LN. In addition, we did not impose harsh N deficiency but instead consistently supplied 0.55 mM N in our LN condition. With this procedure, we gradually induced mild N deficiency in our plants, explaining the rather slow induction of LN-responsive genes. To clarify this point in the text, we now write (lines 326-327): “After 6 days of growth on conditions that evoke mild N deficiency, *BAK1* transcript levels were significantly upregulated (Fig. 7a).”

Line 484 - This selection of similar sized seedlings is biasing the sampling - if there is variation in the growth at early stages due to the mutants, this is nullifying any of the impact of those mutations. Please justify why this approach was taken.

Response: We are sorry for not being clear when describing the procedure in the previous version. Here, “seedlings of similar size” refers to homogeneous seedlings of representative size for each genotype/independent transgenic line. We now clarified this point in lines 500-505: “Homogenous seven-day-old seedlings of representative size for each genotype/independent transgenic line were transferred to new plates with identical sucrose, agar and nutrient composition as described above but supplied with either 11.4 mM (HN) or 0.55 mM (LN) N. If not indicated otherwise, plants were cultivated for 9 days on these conditions.”

Figure 3b - Why are these plants are at a different age than the previous data? The authors’ response to reviewers is justified but this needs to be included in the manuscript.

Response: Thanks for pointing this out. We have now amended the text as follows (lines 553-555): “Following the procedure described earlier for testing BSK-dependent BR responses^{29,30}, we determined BR sensitivity by measuring the length of primary roots and hypocotyls 6 days after germination.”

I would like to see which of the individual lines are statistically different from the quadruple mutant. Also, why are these lines not compared to Col-0?

Response: The aim of this experiment was to compare the ability of each BSK3 variant to stimulate primary root elongation in a less confounding genetic background (i.e., where the absence of four BSKs leads to a robust short root phenotype) and not to test if the expression of a single BSK gene can compensate for the lacking activity of three other closely related homologs. Following the reviewer's suggestion, we carried out a statistical analysis to compare each independent line with the *bsk3,4,7,8* quadruple mutant (see revised Fig. 3b). These results show that primary root elongation was significantly restored in all BSK3-complementing lines, even though to a different extent.

A better presentation of the data would sort the lines by root length measurement to see what the distribution of the lines actually is. This would increase the argument that the P allele lines comprise more of the shorter primary roots measured. The current organization is difficult to interpret.

Response: Thanks for this suggestion. We now sorted the lines by root length and added a dashed line to indicate the average primary root length of all transgenic lines expressing the BSK3-P variant (see revised Fig. 3b). Now, it is possible to recognize that 11 out of 14 independent lines expressing the BSK3-L variant had longer primary roots than the average of all BSK3-P expressing lines.

Figure 4 - B and C - it is unclear which letters are associated with which groups.

Response: To clarify this point, we have now improved the legend of Figure 4 as follows "Numbers in red indicate relative root or hypocotyl length (+BL/-BL). Different letters indicate significant differences for relative root or hypocotyl length at $P < 0.05$ according to one-way ANOVA and post-hoc Tukey test."

Why are all categories not included in the ANOVA and posthoc Tukey testing?

Response: As we described in the Methods (lines 601 to 606), for the analyses of transgenic complementation assays, we employed the Welch's unequal variance t-test on ranked data as proposed by Ruxton (2006), because we had different numbers of individuals for each group and because basic assumptions for ANOVA and Tukey's test were violated (e.g., the absence of equal variance for the values within the groups due to Mendelian segregation within independent T2 lines and construct-dependent effects in different groups).

Figure 4 - G and H need to be analyzed using an ANOVA and post-hoc Tukey test to determine statistical significance between all the genotypes.

Response: As mentioned in our previous response, we applied Welch's unequal variance t-test and carried out pairwise comparisons. We found that Col-0 had significantly longer primary root and faster growth rate than *bsk3,4,7,8* and its complemented lines expressing either BSK3 variant. Both BSK3 variants could significantly rescue primary root growth of *bsk3,4,7,8*, but their primary roots were still shorter than Col-0. This is consistent with the previous finding that enhancing BR signaling inhibits root growth (Gonzalez-Garcia et al., 2011, Development). Notably, overexpressing BSK3-L had a stronger inhibitory effect on root growth compared to BSK3-P. This result supports our conclusion that BSK3-L is more effective than BSK3-P.

The shoot N concentration needs to be added as a supplemental figure to the paper. This data is important as the authors claim that increasing root size will have greater yield, and this suggests that that assumption is not always the case. Not including this data is misleading.

Response: As suggested, we included the data in Supplementary Fig. 13 and described in lines 279-284: “We then tested whether the superior root length brought about by the L-allele also led to elevated N uptake. While we found that under LN supply *bsk3,4,7,8* accumulated significantly less N in the shoot, complementation with either BSK3 variant did not restore shoot N accumulation (Supplementary Fig. 13). This failure may be due to additional, non-redundant functions of BSK proteins and of BRs in the transcriptional regulation of N transporters^{45,46}.”

Minor Comments:

Line 48: In the split root assays there are no primary roots - all the roots studied are lateral roots - secondary or tertiary

Response: Thanks for spotting this. We now amended the text to (lines 48-54): “Nitrate availability and the plant N status can also modulate auxin-dependent primary root and lateral root outgrowth via a regulatory module consisting of miR393 and the auxin receptor AFB3¹². Interestingly, downstream of AFB3, the transcription factor NAC4 and its target OBP4 specifically regulate lateral root initiation and emergence while having no effect on primary root growth¹³. This observation agrees with the finding that individual root traits display a high degree of independence across different N environments¹⁷.”

Line 173: This statement should be altered to state that the transgenic rescue is not statistically significantly similar to WT.

Response: Thanks, we now revised the text to lines 174-177: “Compared to *bsk3*, the elongation of both primary and lateral roots in response to LN could be significantly increased by introducing a genomic fragment containing the promoter and coding regions of BSK3 into the *bsk3* mutant (Supplementary Fig. 6).”

Line 185: For Figure 2 make a plot with total root length - this is addressed in the discussion of this figure but no combined data is provided.

Response: For the sake of space, we show total root length together with lateral root density in Supplementary Figure 7 in the previous version of manuscript (Supplementary Fig. 8 in the newly revised version). In the text, we refer to the results of total root length in lines 198-199: “Consequently, these plants showed no significant increase in total root length under LN supply (Supplementary Fig. 8a).”

Line 278 - Specify that the “root foraging” response is for primary roots only.

Response: As detailed above, although *BSK3* gene mapped by GWAS using the primary root length under mild N deficiency, our phenotypic characterization of *bsk3* and two BSK3 variants revealed that this signaling protein also regulates lateral root elongation in response to low N. Since the elongation of the primary and lateral roots is a major, probably even the dominating component in “root foraging”, we consider it justified to state that BSK3 significantly modulates the root foraging response. However, we agree with the reviewer that at this position in the text, a more detailed statement may be adequate. The revised sentence (lines 288-291) now reads: “Together, these data

provided further support for a specific role of BSK3 in root elongation and demonstrated that the BR sensitivity gained by L319P substitution in BSK3 is critical for the extent of the root elongation in response to LN.”

REVIEWERS' COMMENTS:

Reviewer #5 (Remarks to the Author):

This research and paper are well organized and clearly detail how BSK3 contributes to root responses to LN. The authors have addressed all of my concerns. I look forward to seeing how this paper will contribute to the discussion of the crosstalk and regulatory interactions between nutrients and hormones.